# Measurements of frazil ice flocs in rivers

Chuankang Pei[1], Jiaqi Yang[1], Yuntong She[1], Mark Loewen[1]

[1]Department of Civil and Environmental Engineering, University of Alberta, Edmonton, AB, T6G 1H9, Canada

*Correspondence to*: Mark Loewen (mrloewen@ualberta.ca)

**Abstract.** Frazil floc sizes and concentrations have been investigated in a small number of laboratory studies but no detailed field measurements have been reported previously. In this study, a submersible camera system was deployed a total of eleven times during the principal and residual supercooling phases in the North Saskatchewan, Peace, and Kananaskis Rivers to capture time-series images of frazil ice particles and flocs. Images were processed to accurately identify flocs and to calculate their sizes and concentrations. Key hydraulic and meteorological measurements were collected and air-water heat fluxes were estimated to investigate their influence on floc properties. A lognormal distribution was found to be a good fit for the floc size distribution. The mean floc size ranged from 1.19 to 5.64 mm and the overall mean floc size was 3.80 mm. The mean floc size decreased linearly as the local Reynolds number increased. The average floc number concentration ranged from $1.80 \times 10^{-4}$ to $1.15 \times 10^{-1}$ cm$^{-3}$. The average floc volumetric concentration ranged from $2.05 \times 10^{-7}$ to $4.56 \times 10^{-3}$ and was found to correlate strongly with the fractional height above the river bed. No significant correlations were found between the air-water heat flux and floc properties. Time series analysis showed that during the principal supercooling phase, floc number concentration and mean size increased significantly just prior to peak supercooling and reached a maximum near the end of principal supercooling. During the residual supercooling phase, the mean floc size did not typically vary significantly even 2.5 hours after the residual phase ended and the water temperature increased above zero degrees.

## 1 Introduction

In northern rivers, individual frazil ice particles form when the water is turbulent and supercooled below its freezing point due to heat loss to the atmosphere. These suspended particles are ice crystals that are inherently adhesive in the supercooled water. As they are transported by the turbulent flow, they may collide with each other due to spatially varying particle velocities resulting from differential rising or due to spatially varying flow velocities created by turbulent eddies and boundary shear (Mercier, 1985). Colliding particles may freeze together forming clusters of particles known as frazil flocs in a process called flocculation (Clark and Doering, 2009). Frazil flocs increase in size either by the thermal growth of the crystals and/or by further aggregation of individual frazil ice particles or flocs. Once frazil flocs gain sufficient buoyancy they rise to the water surface forming surface ice pans or are deposited under existing surface ice contributing to their mass increase (Hicks, 2016). In addition, turbulent flow may transport flocs to the river bed where they may adhere to the bed forming anchor ice (Kempema et al., 1993). Once the surface ice pan concentration is high enough, congestion of incoming ice pans will occur at certain

locations where there is a flow constriction and a solid ice cover will form and propagate upstream (Beltaos, 2013). The formation of a continuous solid ice cover insulates the flowing water from further heat loss to the atmosphere, thus preventing the occurrence of supercooling and the production of frazil ice until the ice cover thaws or breaks up (Beltaos, 2013). Frazil flocs may cause serious problems at hydroelectric facilities and water treatment plants by adhering to water intake, trash racks and partially or fully blocking the flow (Ettema and Zabilansky, 2004; Barrette, 2021, Ghobrial et al., 2024). Therefore, it is important to obtain a better understanding of the properties of frazil flocs as well as their evolution to better model and predict their behavior.

As the constructing unit of frazil flocs, individual frazil ice particles have been investigated both in laboratory settings and field. These particles exhibit various forms including dendric, needle, and irregular but are predominately disc-shaped with diameters ranging from 0.022 to 6 mm (McFarlane et al., 2017) and diameter-to-thickness ratios of 11 to 71 (McFarlane et al., 2014). A lognormal distribution can be used to describe the particle size distribution (Daly and Colbeck, 1986; Clark and Doering, 2006; McFarlane et al., 2015). During the principal supercooling period when the water temperature varies transiently, the time from the start of supercooling to when a steady residual supercooling water temperature is reached, the mean diameter of particles was found to first increase before reaching an approximately constant value (Clark and Doering, 2006; McFarlane et al., 2015). At the same time the number concentration of suspended particles first increased slowly then more rapidly, peaking just after peak supercooling occurred (i.e., the minimum water temperature) (McFarlane et al., 2015; Ye, 2002; Clark and Doering, 2006). The rapid increase in particle concentration was attributed to secondary nucleation which refers to the formation of new crystals due to the presence of stable parent crystals (Evans, et al., 1974). After peaking the particle concentration decreased as particles were removed via flocculation.

There have been a small number of laboratory studies that investigated the properties of frazil flocs as well as the flocculation process. Park and Gerard (1984) used artificial flocs fabricated from plastic discs to investigate the hydraulic characteristics of frazil flocs. They found that the sharp-edged floc surface resulted in a significantly higher drag coefficient compared to a solid smooth sphere of the same size and density. Kempema et al. (1993) conducted racetrack flume experiments to investigate interactions of frazil and anchor ice with sediments. They observed that in freshwater frazil easily agglomerated into roughly spherical flocs up to 8 cm in diameter. Flocs that struck the bed tended to entrain sediments into their voids and become heavy and settle to the bottom in the shelter of ripples forming anchor ice. Reimnitz et al. (1993) observed the characteristics and behaviour of rising frazil in seawater using a stirred vertical tube or tank. They found that individual frazil crystals combine rapidly into flocs with diameters as large as 5 cm. The rise velocities of flocs ranged from 1 to 5 cm s$^{-1}$ and rapidly rising large flocs induced small-scale turbulence. The porosities of the resulting surface slush accumulations ranged from 0.68 to 0.85, with an average of 0.77. Clark and Doering (2009) investigated frazil flocculation under different turbulence intensities using a counter-rotating flume. Results showed that higher levels of turbulence increased the rate of secondary nucleation, inhibited the formation of large flocs, and produced more dense flocs.


Schneck et al. (2019) measured the size and number concentration of frazil ice particles and flocs in water of varying salinity
using a stirred frazil ice tank. Results showed that the mean floc size was 2.57 mm in freshwater and 1.47 mm in saline water
and a lognormal distribution fit the floc size distributions closely. The floc porosity was estimated to vary from 0.75 to 0.86.
Time series measurements of floc properties indicated that, in freshwater, the floc number concentration and mean size started
to increase significantly just prior to peak supercooling, reached a maximum shortly afterwards. After that floc number
concentration decreased slowly while the mean floc size continually increased very slowly during the principal supercooling
period.

The above studies were all conducted in laboratory facilities that do not replicate the complex natural environment.
Measurements of frazil flocs in supercooled rivers are needed to verify the laboratory results and improve numerical river ice
process models. However, no detailed quantitative field measurements of the properties or evolution of frazil flocs have been
reported in the literature. The objective of this study was to determine the statistical characteristics and temporal evolution of
floc sizes and concentrations, as well as to investigate the key factors affecting the properties of frazil flocs in rivers. A
submersible high-resolution camera system was used to capture time-series images of frazil flocs. Images were analyzed to
accurately determine floc sizes and concentrations. Key hydraulic and meteorological measurements were collected and air-
water heat fluxes were estimated to investigate their influence on floc properties. Time series of floc size, number concentration
and volumetric concentrations as well as size distributions measured in rivers during the principal and residual supercooling
phase are presented for the first time.
**2 Study reaches**
Measurements were conducted in three regulated Alberta rivers, the North Saskatchewan River (NSR) at Edmonton, the Peace
River (PR) near Fairview, and the Kananaskis River (KR). Figure 1 shows the geographical locations of the study reaches,
deployment sites and weather stations. The characteristics of the study reaches are summarized in Table 1. The turbulent
dissipation rate in Table 1 was estimated using the listed slope as well as the average depth and width following Clark and
Doering (2008). The three rivers are significantly different in terms of their size and hydraulic characteristics. The flow of the
NSR is regulated by the Brazeau and Bighorn Dams which are ~233 km and ~423 km upstream of the Laurier Park site,
respectively. A daily water level fluctuation of 0.3 to 0.4 m occurred in the study reach due to hydropeaking (McFarlane et al.,
2017). The estimated turbulence dissipation rate is 0.0058 $m^2$ $s^{-3}$. Freeze-up typically starts in early November and ends in
early to late December with the formation of a static ice cover. However, the 2022 winter freeze-up progressed in a surprisingly
rapid manner, starting on Nov 5, 2022, and ending just three days later on Nov 8, 2022.

PR has the largest average discharge, depth, and width of the three rivers (Table 1). The estimated turbulence dissipation rate
is 0.0051 $m^2\ s^{-3}$ which is slightly smaller than NSR. The flow of PR is regulated by the W.A.C Bennett Dam and the Peace
Canyon Dam which are ~309 km and ~ 288 km upstream of the Fairview water intake deployment site, respectively. These
outflows at the dams are relatively warm water (~6 °C) during the winter, affecting the river thermal regime for up to 550 km
downstream of the dams (Jasek and Pryse-Phillips, 2015) which is ~250 km downstream of the deployment site. Therefore,
supercooling and frazil ice generation only occurs at the deployment site when the zero-degree isotherm is located upstream
and ceases when it retreats downstream. This unique condition allows freeze-up to persist until the ice front reaches the
Fairview intake site typically in mid-January.

KR is the smallest of the three rivers in terms of average discharge, depth, and width (Table 1). It has the largest turbulence
dissipation rate with a value of 0.2066 $m^2\ s^{-3}$, which is not unexpected since KR is a small-steep river in the mountains. The
flow is regulated by the Pocaterra Dam which is 12 and 31 km upstream of the Fortress and Evan Thomas deployment sites,
respectively. In winter, a dramatic discharge fluctuation from ~1 $m^3\ s^{-1}$ to 21 $m^3\ s^{-1}$ occurred daily in the study reach due to
hydropeaking (Government of Alberta, 2023). Low flows promote border ice formation reducing channel width, while high
flows cause overtopping of existing ice and/or banks and prevent the formation of a complete ice cover. Without an ice cover
to insulate the water, supercooling events and frazil generation occur when the air temperature is sufficiently cold.

**Table 1: Summary of the study reach characteristics.**

| River | Slope | Average discharge ($m^3\,s^{-1}$) | Average depth (m) | Average width (m) | Average $D_{100}$ of suspended sediment (mm) | Estimated turbulence dissipation rate ($m^2\,s^{-3}$) |
|---|---|---|---|---|---|---|
| NSR | 0.00035 | 220 | 1.40 | 136 | 0.50 | 0.0058 |
| PR | 0.00025 | 1586 | 2.56 | 227 | 0.68 | 0.0051 |
| KR | 0.005 | 15 | 0.61 | 32 | N/A | 0.2066 |

*Note*: Slope, average discharge, average depth, and average width were obtained from Kellerhals et al. (1972); Average $D_{100}$
of suspended sediments were computed from Water Survey of Canada historic size distribution data measured at North
Saskatchewan River at Edmonton  (05DF001) and Peace River at Dunvegan Bridge (07FD003) (Water Survey of Canada,

116 2023).

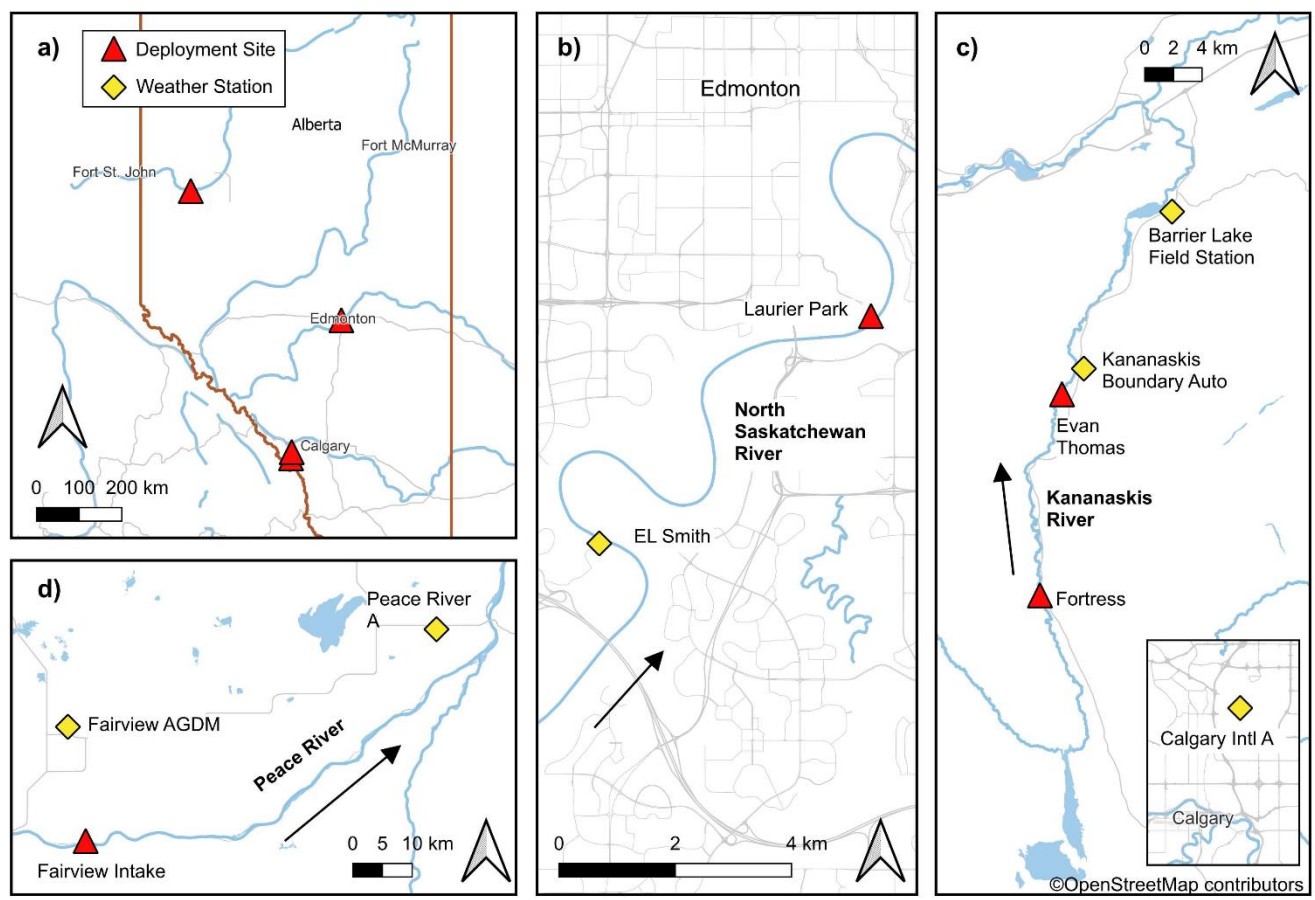


Figure 1: Maps showing (a) the locations of the deployment sites in Alberta, enlarged views of the locations on (b) the North Saskatchewan, (c) Kananaskis, and (d) Peace rivers. This map was produced with QGIS software (https://qgis.org/en/site/) using the data provided by © OpenStreetMap contributors (https://www.openstreetmap.org/copyright) and MapTiler (http://openmaptiles.org/).

## 3 Instrumentation, methodology and deployments

A submersible camera system initially designed for imaging suspended frazil ice particles named "FrazilCam" (McFarlane et al., 2017) was modified in this study to image frazil flocs in the water column. Figure 2 shows the modified configuration of the FrazilCam system. A 36-megapixel Nikon D800 DSLR camera equipped with a Micro-Nikkor 60 mm f/2.8D lens was used to image underwater frazil ice particles and flocs. The camera was enclosed in an Ikelite waterproof housing. Two 16 cm × 16 cm Cavision linear glass cross-polarizing filters were mounted 3.6 cm apart, which is 1.6 times larger than the original configuration. A PVC enclosure with a brass fitting on the top was installed in between the camera lens and polarizing filters to prevent ice or debris from flowing through this region blocking the camera field-of-view (FOV). The brass fitting was used for hot water injection to melt any ice that was initially trapped inside the enclosure. A Nikon SB-910 Speedlight flash in a

Subal SN-910 waterproof housing was used as the light source, and a 5 mm thick white acrylic board was placed in between
the polarizers and flash to diffuse the light. The camera settings were determined by submerging the system in a laboratory
tank filled with tap water and capturing images of a transparent plastic ruler placed inside the camera FOV. This yielded an
ISO of 6400, aperture f/25, and a shutter speed of 1/320. The configuration resulted in an image scale of 25.6 $\mu m$ per pixel
and an average FOV of 11.6 cm by 15.6 cm which is 6 times larger than the original configuration. The reason for enlarging
the FOV and increasing the gap between the polarizers was to enable larger flocs to pass through and fit within the FOV.
At the start of each deployment, the camera was programmed to acquire 5 images at 1 Hz every 9 s, 15 s, or 18 s depending
on the field conditions until the battery was depleted. A longer sampling interval (e.g. 18 s) was chosen for some deployments
to prolong the deployment duration with the goal of capturing a complete supercooling event. Just prior to deployment of the
FrazilCam in the river, the polarizers were rinsed with hot saline water to prevent ice crystals from forming on them once
submerged. The system was then quickly deployed in the river and the PVC enclosure was filled with hot fresh water from an
elevated container. During deployments, anchor ice often formed on system components as shown in Fig. 3 and ice that formed
on the polarizers could obstruct the FOV of the camera. To prevent or mitigate this problem, the polarizers were inspected
every 30 to 60 minutes and hot saline water was injected onto the polarizers to melt any ice crystals.
During each deployment, an RBR Solo T (accuracy $\pm\, 0.002°C$) temperature logger sampling every second was attached to the
top of the frame to measure water temperature, and a Van Essen Diver (accuracy $\pm\, 1$ cmH$_2$O) water level logger sampling
every 10 minutes was attached to the bottom of the frame to measure the water depth (Fig. 2). The water depth during the PR
deployments was measured using a wading rod since the Diver stopped working at that time. For all deployments the depth-
averaged water velocity was estimated using velocities measured adjacent to FrazilCam at 60% of the water depth. During the
2021 winter, the water velocity was measured using a 2-MHz Nortek AquaDopp High Resolution Acoustic Doppler Current
Profiler sampling every second with a blanking distance of 0.1 m and averaging every two minutes. For the rest of the
deployments, the water velocity was measured using a SonTek Flow Tracker Handheld Acoustic Doppler Velocimeter (ADV)
sampling every second for a total duration of 50 seconds.

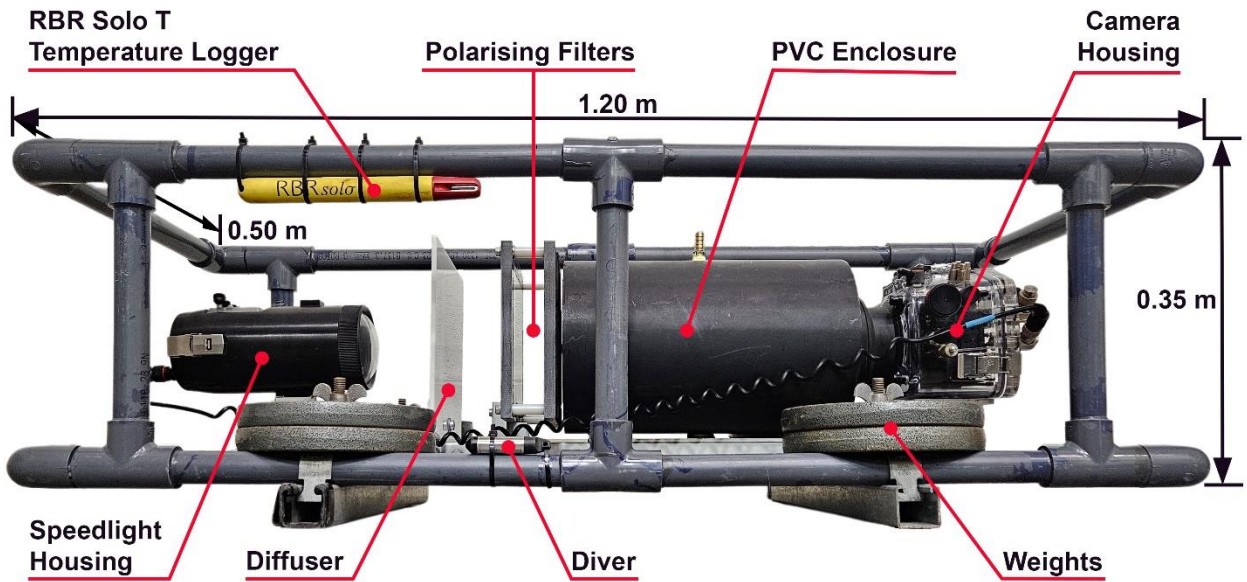

Figure 2: An image showing the configuration of the FrazilCam system.

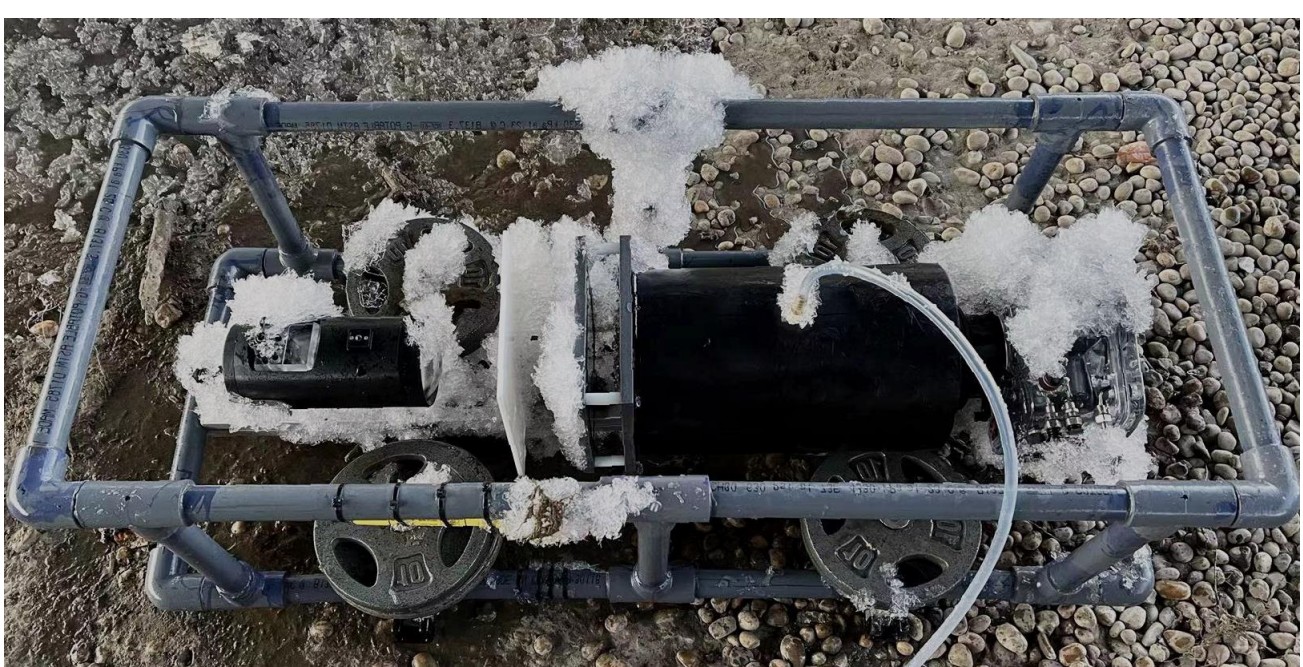

Figure 3: An image showing the ice accumulation on the FrazilCam system.

Meteorological conditions for the NSR reach were measured by a weather station installed at the E.L. Smith water treatment plant, which is located ~90 m from the river bank and ~6 km upstream of Laurier Park site (Fig. 1b). The weather station measures the air temperature, solar radiation, relative humidity, atmospheric pressure, wind speed and direction every minute

and logs data every 10 minutes. An Apogee SN-500-SS net radiometer was deployed on the river bank at this location, measuring incoming and outgoing shortwave/longwave radiation every minute and logging data every 10 minutes. For the PR, 1-hour interval meteorological data were obtained from ECCC station Fairview AGDM (ID: 3072525) and 3-hour interval cloud coverage data was obtained from the closest ECCC station Peace River A (ID: 3075041) as shown in Fig. 1d. For the KR, the Kananaskis Boundary Auto weather station operated by Alberta Forestry, Parks and Tourism (ACIS, 2023) was used to obtain 1-hour interval air temperature, humidity, wind speed, and wind direction data. In addition, 1-hour solar radiation data was obtained from the University of Calgary Barrier Lake Field Station weather station (University of Calgary, 2023), and 3-hour cloud coverage data was obtained from the closest ECCC station Calgary Intl A (ID: 3031092) as shown in Fig. 1c. Table 2 summarizes the distance between weather stations and deployment sites. All weather stations are located within 30 km of their nearby deployment sites, except for those providing cloud coverage data for PR and KR.

**Table 2: The distances between weather stations and deployment sites.**

| River | Deployment site | Distance - weather station |
|---|---|---|
| NSR | Laurier Park | 6 km - E.L. Smith |
| PR | Fairview Intake | 18 km - Fairview AGDM; 68 km - Peace River A |
| KR | Evan Thomas | 2 km - Kananaskis Boundary Auto; 15 km - Barrier Lake Field Station; 82 km - Calgary Intl A |
| | Fortress | 16 km - Kananaskis Boundary Auto; 28 km - Barrier Lake Field Station; 88 km - Calgary Intl A |

The FrazilCam system was deployed a total of eleven times during the 2021 and 2022 freeze-up periods, images of the FrazilCam during two of the deployments are shown in Fig. 4. The image sampling protocols were 5 images at 1 Hz every 9 s for all NSR and KR-E1 deployments, for KR-F1 and KR-F2 5 images at 1 Hz every 15 s, and for all PR deployments 5 images at 1 Hz every 18 s. Table 3 lists the detailed location, date, time, number of images processed, and deployment number for each deployment. The mean air temperature $\overline{T_a}$, mean water depth $\bar{d}$, depth-averaged flow velocity $\overline{U}$, and the local Reynolds number $Re$ computed from $\bar{d}$ and $\overline{U}$ are also presented in Table 3. Eight of eleven deployments started in the afternoon around 2 PM ~ 7 PM when the effect of solar radiation reduced decreasing heat gain of the water body, the time duration of deployments ranged from 1:48 to 3:21. As can be seen from Table 3, during these deployments $\overline{T_a}$ ranged from -3.5 °C to -20.6 °C, $\bar{d}$ ranged from 0.41 m to 1.24 m, $\overline{U}$ ranged from 0.12 m s$^{-1}$ to 0.36 m s$^{-1}$, and $Re$ ranged from 44866 to 160714, respectively, indicating that frazil floc properties and concentrations were measured and analyzed over a wide range of meteorological and hydraulic conditions. The eleven deployments captured various phases of supercooling but NSR-L4 was the only deployment that captured a complete principal supercooling phase (i.e., from when the water temperature first dropped below zero to when an approximately stable residual temperature was reached).

Table 3: Summary of the FrazilCam deployments and site conditions including the number (#) of images captured, mean air temperature $\overline{T_a}$, mean water depth $\bar{d}$, depth averaged water velocity $\overline{U}$, and local Reynolds number $Re$.

| River | Date (yyyy.mm.dd) | Time period (hh:mm~hh:mm) | # of processed images | Site | Deployment No. | $\overline{T_a}$ (°C) | $\bar{d}$ (m) | $\overline{U}$ (m s$^{-1}$) | $Re$ |
|---|---|---|---|---|---|---|---|---|---|
| NSR | 2021.12.3 | 16:41~18:49 | 4,099 | Laurier Park | NSR-L1 | -7.2 | 0.89 | 0.21 | 104,297 |
| | | 19:05~21:34 | 4,797 | Laurier Park | NSR-L2 | -10.5 | 0.84 | 0.17 | 79,688 |
| | 2021.12.9 | 14:46~17:09 | 4,688 | Laurier Park | NSR-L3 | -3.5 | 1.24 | 0.19 | 131,473 |
| | 2021.12.12 | 15:02~16:50 | 3,495 | Laurier Park | NSR-L4 | -4.6 | 0.87 | 0.22 | 106,808 |
| | | 17:08~19:31 | 4,091 | Laurier Park | NSR-L5 | -9.2 | 0.86 | 0.20 | 95,982 |
| | 2022.11.7 | 14:31~16:22 | 3,596 | Laurier Park | NSR-L6 | -12.1 | 0.80 | 0.36 | 160,714 |
| PR | 2022.12.12 | 10:40~13:57 | 3,155 | Fairview Intake | PR-F1 | -20.6 | 0.82 | 0.30 | 137,277 |
| | 2022.12.13 | 9:41~13:02 | 3,208 | Fairview Intake | PR-F2 | -6.0 | 0.74 | 0.23 | 94,978 |
| KR | 2023.1.29 | 18:00~20:02 | 3,728 | Evan Thomas | KR-E1 | -15.8 | 0.41 | 0.22 | 50,335 |
| | 2023.1.30 | 14:46~17:59 | 3,379 | Fortress | KR-F1 | -11.1 | 0.55 | 0.30 | 92,076 |
| | 2023.1.31 | 7:28~10:39 | 3,610 | Fortress | KR-F2 | -13.3 | 0.67 | 0.12 | 44,866 |

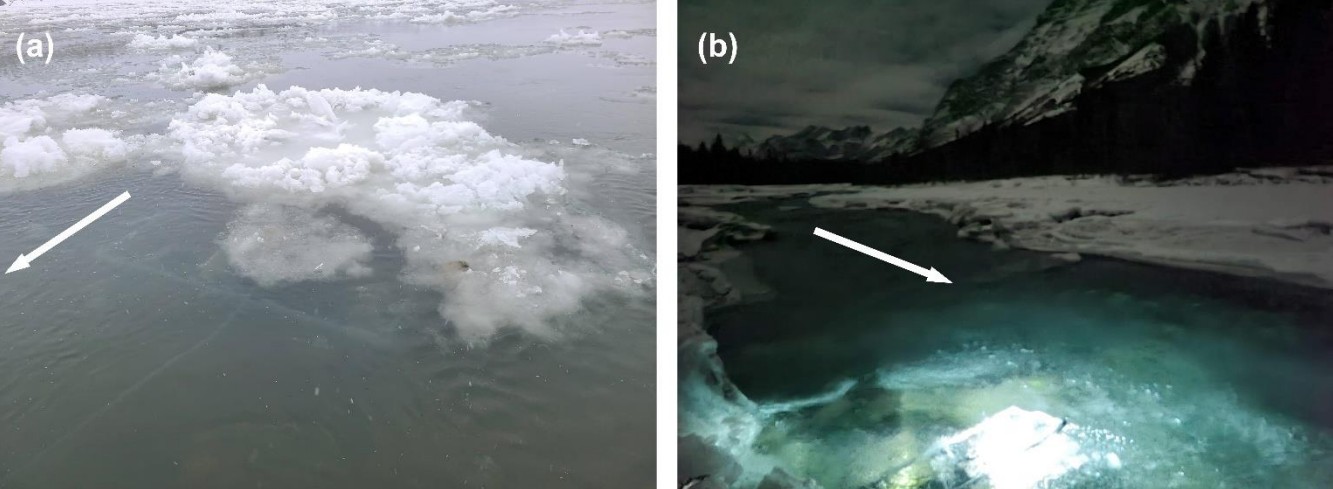

Figure 4: Image of the FrazilCam deployed during (a) NSR-L6, and (b) KR-E1. The arrow indicates the flow direction.

## 4 Data processing

### 4.1 Image processing

Figure 5a shows an example of a raw FrazilCam image with individual frazil ice particles, flocs, and ice crystals frozen on the polarizer. Frazil ice particles are predominantly disk-shaped (McFarlane et al., 2017) and therefore depending on their orientation appear in the images as shapes that vary from a line to a circle with the majority being ellipses. Flocs form through the aggregation of frazil ice particles, resulting in varying shapes depending on the number, shape, and size of attached particles. Ice crystals sometimes attached and froze to the surface of the polarizers despite the periodic hot saline water rinsing. These crystals may appear anywhere in the image, blocking certain regions of the FOV.

Figure 6 shows a flow chart of the image processing procedure used for extracting frazil floc properties. For each deployment, images were first manually inspected to exclude those taken when the polarizers were being rinsed, which constitutes 2 ~ 14% of the total images captured. Each image was then processed using an iterative thresholding algorithm developed by McFarlane et al. (2014) to determine the location and extent of each object. Objects intersecting with the image boundary were eliminated, which also removed the ice crystals that were frozen near polarizer edges. For frozen ice crystals that did not intersect with the image boundary, the affected image area was removed either by cropping or masking, or a combination of both (Fig. 6). The corresponding processed binary image is shown in Fig. 5b.

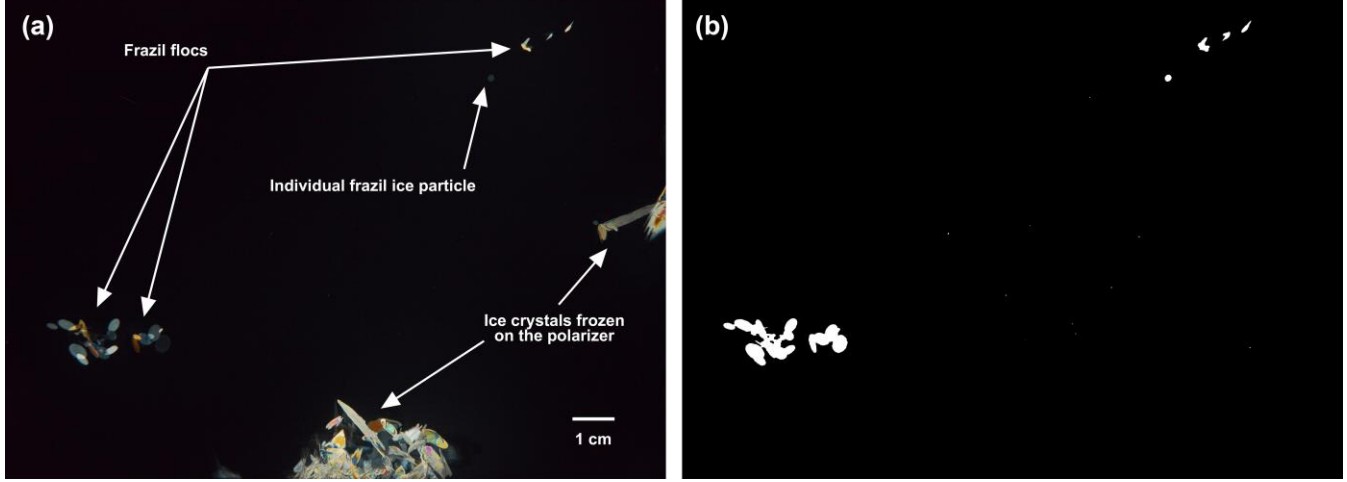

**Figure 5: An example of (a) a raw FrazilCam image captured on Dec 3, 2021, and (b) the corresponding processed binary image.**

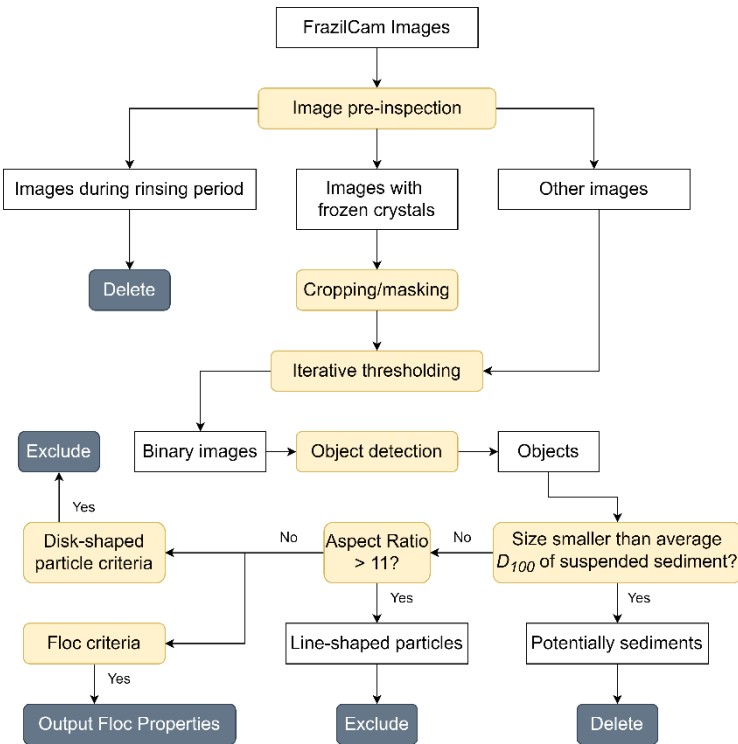

**Figure 6: A flow chart showing the image processing procedure.**

The processed binary images were analyzed to compute each object's basic geometric characteristics such as area, perimeter, centroid, as well as the major and minor axis length of its fitted ellipse. The size $S$ of both frazil particles and flocs was defined as the major axis length of its fitted ellipse (Clark and Doering, 2009). The objects in the processed images may include small-suspended sediments that were thin enough to refract light, which may significantly distort the size distribution of frazil ice particles and flocs (McFarlane et al., 2017; Pei et al., 2022). McFarlane et al. (2019a) used a support vector machine (SVM) to distinguish between ice particles and sediments and compute accurate particle size distributions. However, this method requires ice-free sediment images at each site for site-specific SVM training, which is not possible for this study due to the lack of ice-free images at the PR and KR sites. Since this study focuses on flocs, which are considerably larger than particles, a simple cut-off criterion was used to minimize the effect of sediment particles in the images. Objects smaller than the average $D_{100}$ of suspended sediment (see Table 1) in a given study reach were removed from the dataset (Fig. 6). For the KR, since no suspended sediment size distribution measurements were available in the literature, the cut-off size was determined to be 0.27 mm, which is twice the average of seven mean sediment size measurements estimated from FrazilCam images by McFarlane et al. (2019b).

For each object, the following geometric parameters were used to classify the objects into either flocs or particles: the ratio of
the object area to that of the fitted ellipse $a/a_e$, the absolute percentage difference between the object perimeter and its fitted
ellipse perimeter $P_{diff\%}$, the ratio of an object's fitted ellipse area to its ellipse perimeter divided by the ratio of the object's
actual area to its perimeter $(\frac{a_e}{P_e})/(\frac{a}{P})$ (McFarlane et al., 2014; Schneck, 2018). Preliminary experiments found that flocs formed
by a very small particle attaching to a significantly larger particle remain approximately elliptical since the boundary does not
change significantly. As a result, comparing changes in overall area/perimeter with the fitted ellipse did not help with
classification. Therefore, the form index was introduced to assess minor changes in object shape (Masad et al., 2001; Al-
Rousan et al., 2007). The form index is calculated using the following equation:
$FI = \sum_{\theta=0}^{\theta=360-\Delta\theta} \frac{|R_{\theta+\Delta\theta}-R_\theta|}{R_\theta}$ , (1)
where $\theta$ is the directional angle and $R$ is the radial length between the centroid of the particle and the boundary of the particle.
The incremental change in angle $\Delta\theta$ is set to 2.81 °, dividing the particle boundary into 128 segments to factor in minor
boundary changes. A perfectly circular object has an $FI$ of 0, and $FI$ will increase as an object's boundary becomes more
irregular.

A total of 568 objects were manually labelled as either flocs (109) or disk-shaped frazil particles (459) to construct a test
dataset to determine the optimal classification criteria of the aforementioned parameters. Results showed that
$\{a/a_e \geq 0.9 \ and \ P_{diff\%} \leq 0.1 \ and \ S \leq 6\}$ for disk-shaped particles, and $\{(a/a_e < 0.9 \ or \ P_{diff\%} > 0.15) \ and \ (\frac{a_e}{P_e})/(\frac{a}{P}) >$
$1.1 \ and \ FI \geq 6\}$ for flocs provided the optimum classification accuracy of 97.0% and 92.7% for particles and flocs,
respectively. In NSR-L4 the camera lens was slightly out of focus due to an accidental jarring of the camera during deployment.
However, because this was the only deployment that captured a complete principal supercooling event, additional processing
was performed on these images to allow for their inclusion in the dataset. Visual examination and analysis of these images
indicated that the blurriness predominantly affected the boundary clarity of dim objects with a mean pixel intensity less than
24 and did not significantly affect brighter objects. Therefore, an additional criterion was introduced for NSR-L4 eliminating
flocs with a mean pixel intensity less than 24. The rate of floc detection in the blurry images from deployment NSR-L4 was
4.1 flocs per minute and it was 4.4 flocs per minute in NSR-L5 which occurred immediately afterwards. Therefore, the
additional criterion, applied to the blurry images, only slightly reduced the number of flocs detected.

In order to prevent line-shaped frazil ice particles from being misidentified as flocs, frazil particles in the shape of a line were
first identified if the aspect ratio of the object (i.e., the ratio between the major and minor axis length) was greater than 11
based on minimum frazil ice particle aspect ratio measurements made by McFarlane et al. (2014) as shown in Fig. 6. Then the
classification criteria mentioned above were applied to the remaining objects to identify disk-shaped particles and flocs (Fig.
6). After classification, the number of flocs $N_T$, mean floc size $\overline{\mu_f}$, standard deviation $\sigma_f$, 95$^{th}$ percentile of floc size $S_{f95,}$
maximum floc size $S_{fmax,}$ average floc number concentration $\overline{C_{fn}}$, and average volumetric concentration $\overline{C_{fv}}$ for each
deployment were computed. It is worth noting that the properties of frazil ice particles were not included in this study since
the cut-off size likely eliminated up to 50% of the particle population which would significantly skew the data. In addition, the
mean floc size $\mu_f$, floc number concentration $C_{fn}$, floc volumetric concentration $C_{fv}$ were computed for each image throughout
a deployment, and a moving average over a period of 35 images was applied to the resulting time series to smooth the data.
Note that the 35-image moving average was computed only if two or more non-zero values occurred in the window, if there
were less than two non-zero values no average value was recorded. This created gaps in the moving average time series and
the rationale for this is that two or more samples are required to compute a valid average value. The measuring volume used
for the concentration calculations was the image FOV times the gap distance between the two polarizers. The volume of a
frazil floc was assumed to be the volume of an ellipsoid with semi-axis lengths $a$, $b$, and $c$ where $a$ and $b$ were equal to the
semi-major and semi-minor axis lengths of the floc's fitted ellipse, and $c$ was equal to the average of $a$ and $b$ but no larger than
the gap between the two polarizing filters. The volume of ice in a frazil floc $V_f$ was estimated as:
$V_f = \frac{4}{3}\pi abc(1 - \eta)$ ,            (2)
where $\eta$ is the porosity of floc taken to be 0.8 (Schneck et al., 2019).

### 4.2 Heat flux analysis at the water surface

The net heat flux $Q_n$ at the river surface is given by:
$Q_n = Q_{sw} + Q_{lw} + Q_E + Q_H$ ,            (3)
where $Q_{sw}$ is the net shortwave radiation; $Q_{lw}$ is the net longwave radiation; $Q_E$ is the latent heat flux; $Q_H$ is the sensible heat
flux. A positive sign denotes heat loss from the surface. $Q_{sw}$ was calculated as:
$Q_{sw} = -(1 - \alpha_{ws})Q_s$ ,            (4)
where $Q_s$ is the measured incoming solar radiation; $\alpha_{ws}$ is the albedo of water surface to solar radiation, taken to be 0.15 for
this study following Howley (2021). The net longwave radiation $Q_{lw}$ was calculated as:
$Q_{lw} = Q_{lw}^{out} - (1 - \alpha_{wl})Q_{lw}^{in}$ ,            (5)
$Q_{lw}^{out} = \varepsilon_w \sigma_{sb} T_{wk}^4$ ,            (6)
where $Q_{lw}^{out}$ is the outgoing longwave radiation emitted from the water; $\alpha_{wl}$ is the albedo of water surface to longwave
radiation, taken as 0.03 (Raphael, 1962); $\varepsilon_w$ is the emissivity of water taken as 0.97 (Ashton, 2013); $\sigma_{sb}$ is the Stefan-
Blotzmann constant ($5.67\times 10^{-8}$ W m$^{-2}$ K$^{-4}$); $T_{wk}$ is the water surface temperature in K. Note that it was assumed that the water
column was completely mixed and therefore the water temperatures that were measured at the top of the FrazilCam frame (i.e.,
not at the water surface) were used in Eq. (6). $Q_{lw}^{in}$ is the incoming longwave radiation which was measured by a net radiometer
for the NSR. For KR and PR, $Q_{lw}^{in}$ is estimated using the following equations:
$$Q_{lw\_c}^{in} = \varepsilon_{ac}\sigma_{sb}T_{ak}^4 \,, \tag{7}$$
$$\varepsilon_{ac} = 1.08[1 - \exp(-e_a{}^{T_{ak}/2016})] \,, \tag{8}$$
$$e_s = 6.11\exp\left(\frac{17.62T_a}{243.12+T_a}\right) \,, \tag{9}$$
$$e_a = RH \times e_s \,, \tag{10}$$
$$Q_{lw}^{in} = Q_{lw\_c}^{in}(1 - N^4) + 0.952N^4\sigma_{sb}T_{ak}^4 \,, \tag{11}$$
where $Q_{lw\_c}^{in}$ is the incoming longwave radiation under the clear sky; $\varepsilon_{ac}$ is the clear sky atmospheric emissivity calculated
using Eq. (8) by Satterlund (1979); $T_{ak}$ is the air temperature in K; $e_s$ and $e_a$ are the saturated and actual vapour pressure of
water, respectively; $RH$ is the relative humidity; $T_a$ is the air temperature in degree Celsius; $N$ is the fractional cloud cover.
Note that Eq. (11) was developed by Konzelmann et al. (1994).

$Q_E$ was calculated using the equation suggested by Ryan et al. (1974) following Yang et al. (2023):
$$Q_E = \left[2.70\left(\frac{T_{wk}}{1-0.378(e_s/P)} - \frac{T_{ak}}{1-0.378(e_a/P)}\right)^{\frac{1}{3}} + 3.2V\right](e_s - e_a) \,, \tag{12}$$
where $P$ is the atmospheric pressure; $V$ is the wind speed. $Q_H$ was calculated from $Q_E$ using Bowen's ratio $B$ as follows:
$$B = \frac{C_aP}{0.622l_v} \times \frac{T_s-T_a}{e_s-e_a} \,, \tag{13}$$
$$Q_H = BQ_E \,, \tag{14}$$
where $C_a$ is the specific heat of air; $l_v$ is the latent heat of vaporization; $T_s$ is the surface water temperature. In a previous study,
Yang et al. (2023) investigated various formulas used to calculate incoming longwave radiation and the latent and sensible
heat fluxes during freeze-up on the North Saskatchewan River in Alberta, and the combination of formulas (Eqs. 7~14) used
in this study were the ones that provided the most accurate results in Yang et al (2023). It is also worth noting that only hourly
meteorological data were available for the KR and PR regions as described in Sect. 3. As a result, the heat fluxes were
calculated on a 1-hour time interval for the KR and PR deployments, and for all the NSR deployments the heat fluxes were
calculated on a 10-min time interval.

## 5 Results

### 5.1 Floc shape, size and concentration

In Fig. 7 images of typical shapes of frazil flocs observed during the different field deployments are presented. Flocs from NSR deployments (Figs. 7a~b) were comprised predominantly of disc-shaped frazil ice particles of varying sizes frozen together. The floc shown in Fig. 7b is representative of flocs observed during deployments NSR-L3 and NSR-L6. As can be seen, it was comprised of much smaller individual particles than the flocs observed during the rest of the NSR deployments (Fig. 7a). Flocs from deployment PR-F1 (Fig. 7c) were comprised of disc-shaped particles, irregular particles, and some needle-shaped particles. Flocs from deployment KR-E1 (Fig. 7d) were formed primarily by densely aggregated irregular particles and some small disc-shaped particles. Flocs from deployments PR-F2, KR-F1 (Fig. 7e), and KR-F2 (Fig. 7f) were mostly comprised of disc-shaped and irregular particles, images of flocs from PR-F2 were not shown since they are similar to those shown in Figs. 7e-f.

Table 4 presents the number of flocs $N_T$, mean size $\overline{\mu_f}$, standard deviation $\sigma_f$, 95th percentile and maximum of the floc size $S_f$, average floc number concentration $\overline{C_{fn}}$, and average volumetric concentration $\overline{C_{fv}}$ for each deployment. The supercooling phase, the minimum water temperature $T_p$, and average net surface heat flux $\overline{Q_n}$ are also presented. Deployments NSR-L1, NSR-L3, and NSR-L4 captured the principal supercooling phase (Principal), while the rest captured only the residual supercooling phase (Residual). $T_p$ ranged from -0.021 °C to -0.031 °C for Principal deployments, and from -0.007 °C to -0.017 °C for Residual deployments. In all deployments $\overline{Q_n}$ was positive indicating an overall heat loss. $N_T$ varied significantly ranging from 442 to 187,288 with the largest $N_T$ of 187,288 occurring during deployment KR-E1. The mean floc size $\overline{\mu_f}$ ranged from 1.19 to 5.64 mm with an overall average of 3.8 mm and $\sigma_f$ ranged from 0.88 to 5.03 mm. $S_{f95}$ was greater than ~8 mm except for deployments NSR-L3 and NSR-L6 with values of 4.44 mm and 2.47 mm, respectively. The largest value of $S_{fmax}$, 99.69 mm, was observed during KR-E1 which also had the largest number of flocs. The average floc number concentration $\overline{C_{fn}}$ varied by three orders of magnitude from $1.80\times10^{-4}$ to $1.15\times10^{-1}$ cm$^{-3}$, and the average floc volumetric concentration $\overline{C_{fv}}$ over four orders of magnitude from $2.05\times10^{-7}$ to $4.56\times10^{-3}$.

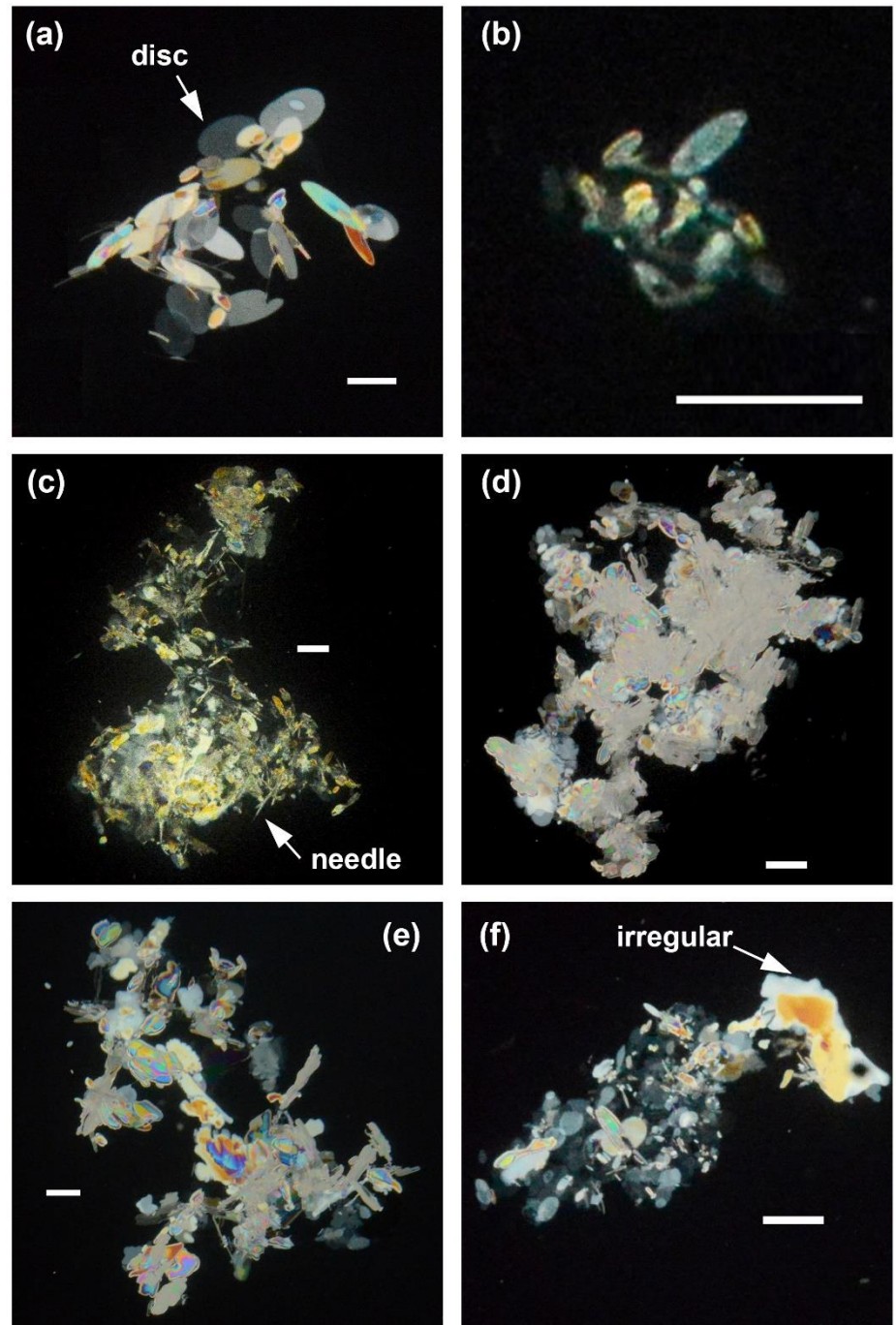

339

Figure 7: Images of frazil flocs of different sizes and shapes from the following deployments: (a) NSR-L1, (b) NSR-L6, (c) PR-F1, (d) KR-E1, (e) KR-F1, and (f) KR-F2. The white scale bar in each image represents a length of 3 mm. Note that in some images the surrounding ice particles were masked out to highlight the floc at the centre of the image.

Table 4: Supercooling phase, minimum water temperature $T_p$, mean net surface heat flux $\overline{Q_n}$, number of flocs $N_T$, mean floc size $\overline{\mu_f}$, standard deviation $\sigma_f$, 95th percentile of floc size $S_{f95}$, maximum floc size $S_{fmax}$, average floc number concentration $\overline{C_{fn}}$, and average volumetric concentration $\overline{C_{fv}}$ for each deployment.

| Deployment No. | Supercooling phase | $T_p$ (°C) | $\overline{Q_n}$ (W m$^{-2}$) | $N_T$ | $\overline{\mu_f}$ (mm) | $\sigma_f$ (mm) | $S_{f95}$ (mm) | $S_{fmax}$ (mm) | $\overline{C_{fn}}$ (cm$^{-3}$) | $\overline{C_{fv}}$ (cm$^3$ cm$^{-3}$) |
|---|---|---|---|---|---|---|---|---|---|---|
| NSR-L1 | Principal | -0.021 | 183.3 | 2,428 | 4.33 | 3.08 | 8.73 | 89.58 | $9.65\times10^{-4}$ | $1.39\times10^{-5}$ |
| NSR-L2 | Residual | -0.009 | 199.5 | 879 | 3.70 | 2.31 | 7.54 | 24.05 | $2.72\times10^{-4}$ | $1.39\times10^{-6}$ |
| NSR-L3 | Principal | -0.023 | 95.4 | 839 | 1.87 | 1.31 | 4.44 | 9.02 | $3.06\times10^{-4}$ | $2.05\times10^{-7}$ |
| NSR-L4 | Principal | -0.031 | 110.3 | 442 | 4.50 | 2.45 | 8.37 | 18.53 | $1.80\times10^{-4}$ | $1.21\times10^{-6}$ |
| NSR-L5 | Residual | -0.016 | 121.8 | 631 | 3.50 | 2.57 | 8.40 | 14.31 | $2.60\times10^{-4}$ | $1.19\times10^{-6}$ |
| NSR-L6 | Residual | -0.017 | 157.5 | 143,097 | 1.19 | 0.88 | 2.47 | 47.16 | $6.75\times10^{-2}$ | $2.99\times10^{-5}$ |
| PR-F1 | Residual | -0.009 | 318.8 | 2,250 | 3.43 | 3.72 | 9.16 | 53.35 | $1.11\times10^{-3}$ | $1.84\times10^{-5}$ |
| PR-F2 | Residual | -0.007 | 107.4 | 1,247 | 4.25 | 5.03 | 13.60 | 53.81 | $5.63\times10^{-4}$ | $1.68\times10^{-5}$ |
| KR-E1 | Residual | -0.008 | 243.3 | 187,288 | 5.64 | 4.79 | 14.28 | 99.69 | $1.15\times10^{-1}$ | $4.56\times10^{-3}$ |
| KR-F1 | Residual | -0.010 | 122.4 | 23,670 | 4.43 | 3.86 | 10.69 | 81.38 | $1.05\times10^{-2}$ | $2.32\times10^{-4}$ |
| KR-F2 | Residual | -0.011 | 275.2 | 15,151 | 4.69 | 4.08 | 11.89 | 68.37 | $6.62\times10^{-3}$ | $1.59\times10^{-4}$ |

## 5.2 Floc size distribution

In Fig. 8, plots of the frazil floc size distribution as well as fitted lognormal distribution curves for four deployments are presented. All of the size distributions obtained from NSR deployments closely resemble deployment NSR-L1 shown in Fig. 8a, except for deployment NSR-L6 shown in Fig. 8b. Size distributions from the KR and PR are well represented by deployments KR-F1 and PR-F1 which are shown in Fig. 8c and Fig. 8d, respectively. It can be seen from Fig. 8 that a theoretical lognormal distribution is a reasonable fit to all of the size distributions but a particularly good fit for deployment KR-F1. This may be attributed to the order-of-magnitude larger sample size for KR-F1 (23,670) compared to NSR-L1 (2,428) and PR-F1 (2,250). The size distribution for NSR-L6 shown in Fig. 8b has the most flocs of the four deployments plotted with a sample size of 143, 097 but it does not fit a lognormal distribution as closely as the others. This is because the distribution was cut off at 0.5 mm to eliminate sediment particles. A similar condition can also be observed for PR-F1 shown in Fig. 8d where the cut-off was 0.68 mm. Note that the cut-offs were applied to all size distributions but only impacted the distribution significantly if there were a significant number of smaller flocs detected.

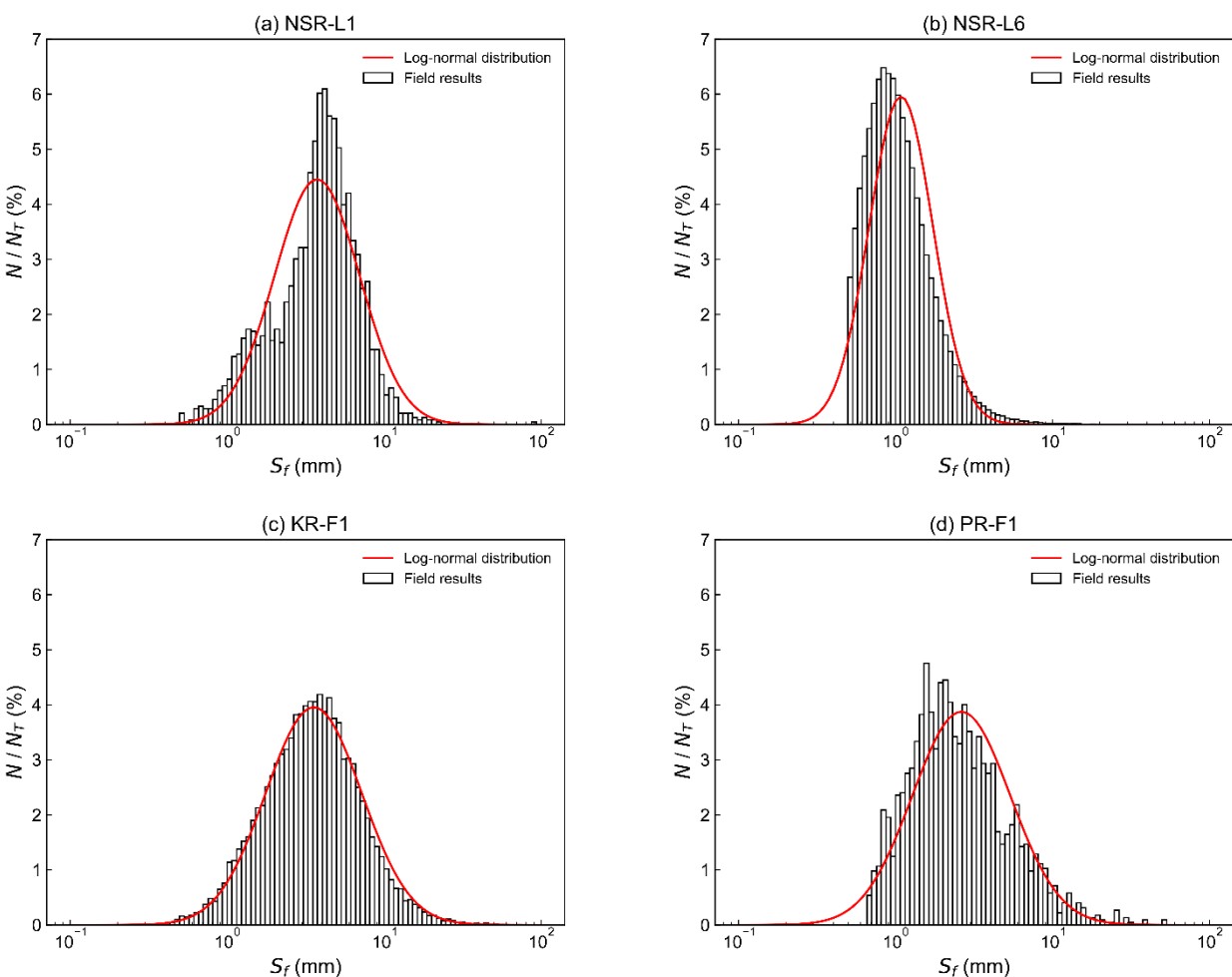

361

Figure 8. Distributions of floc size $S_f$ for deployments (a) NSR-L1, (b) NSR-L6, (c) KR-F1, and (d) PR-F1. The red line denotes a fitted lognormal distribution, $N$ is the number of flocs in each bin, and $N_T$ is the total number of flocs.

**5.3 Time series**

Time series plots of water and air temperatures $T_w$ and $T_a$, heat flux $Q$, floc mean size $\mu_f$, floc number concentration $C_{fn}$, and floc volumetric concentration $C_{fv}$ for deployments NSR-L4, KR-F1, and PR-F2 are presented in Figs. 9, 10 and 11, respectively (Note that similar time series plots for the other eight deployments are presented as Figs. S1-S8 in the Supplement). Deployment NSR-L4 occurred during the principal supercooling phase and is the only deployment that captured the entire principal supercooling phase, while KR-F1 and PR-F2 captured the middle and end of the residual supercooling phase, respectively.

During NSR-L4 (Fig. 9a) supercooling started at 15:25 and after that $T_w$ decreased almost linearly at a cooling rate of -0.0009 °C min$^{-1}$, reached a $T_p$ of -0.031 °C (i.e., peak supercooling) at 16:02 and then started to increase and reached a stable residual temperature of -0.010 °C at 16:37. $T_a$ decreased from -1.7 to -7.2 °C with an average of -4.6 °C. Figure 9b shows that the net heat flux $Q_n$ increased from 26 W m$^{-2}$ to 150 W m$^{-2}$ primarily due to the decrease in the magnitude of shortwave radiation $Q_{sw}$. The rest of the heat flux components remained positive (heat loss) and relatively stable throughout the deployment, with $Q_{lw}$ being the dominant component. In Fig. 9c $\mu_f$ began increasing significantly ~7 minutes before the peak supercooling temperature was reached, reaching a maximum of 7.8 mm ~37 minutes after peak supercooling, then it decreased to ~6 mm and remained approximately constant afterwards. Figure 9d shows that significant numbers of frazil particles were detected ~15 minutes before peak supercooling with $C_{fn}$ values below $2 \times 10^{-4}$ cm$^{-3}$. At ~ 2 minutes before peak supercooling $C_{fn}$ increased rapidly and peaked ~30 minutes after peak supercooling at a value of $9.3 \times 10^{-4}$ cm$^{-3}$ and then decreased to $2 \times 10^{-4}$ cm$^{-3}$ at the end of the deployment. Figure 9e shows that $C_{fv}$ only increased notably after peak supercooling and reached a value of $8.8 \times 10^{-6}$ ~30 minutes after the peak supercooling. After that it decreased before spiking to $1.6 \times 10^{-5}$ ~38 minutes after the peak supercooling and then decreased to $1.7 \times 10^{-6}$ at the end. An examination of the images showed that the spike was caused by several large flocs up to 18.5 mm in size.

During KR-F1, $T_w$ fluctuating continuously around -0.008 °C, except for one anomalous spike that occurred at 17:03 (Fig. 10a), which was caused by ice contacting the sensor when the polarizers were being rinsed. Additionally, periodic upward spikes with a period of 1 minute and magnitude of ~0.001 °C were visible on the plot. While the cause of these spikes remains uncertain, it is worth noting that their magnitude falls within the range of accuracy of the sensor. The air temperature was relatively stable with $T_a$ varying between -10 to -12 °C. In Fig. 10b, $Q_n$ rose during the deployment from -2 W m$^{-2}$ to 261 W m$^{-2}$ largely due to the decrease in the magnitude of $Q_{sw}$. Note that the heat flux components here were computed on a 1-hour time interval. In Figs. 10c-e, there are gaps in the data during these time periods 15:33 ~ 15:38, 16:17 ~ 16:23, 16:58 ~ 17:04, and 17:34 ~ 17:39, that are visible as short time series segments with zero slope. These were created when the images collected during the time periods when the polarizers were being rinsed were removed from the dataset. In Fig. 10c, $\mu_f$ fluctuated around ~ 4 mm before significantly increasing at 17:40, eventually reaching 5.9 mm by the end of the deployment. Similar trends are evident in Figs. 10d-e for $C_{fn}$ and $C_{fv}$, respectively. At 17:41 $C_{fn}$ started to increase significantly and reached a peak value of $4.5 \times 10^{-2}$ cm$^{-3}$ at 17:53 while $C_{fv}$ started to increase significantly at 17:50 and eventually peaked at a value of $2.8 \times 10^{-3}$. A hydropeaking wave arrived at the Fortress site at 17:25 increasing the depth by 19% by the end of the deployment and causing rapid increases in floc size and concentration.

During deployment PR-F2, $T_w$ was initially at -0.006 °C but then increased above zero at 10:21, and eventually reached 0.033 °C at the end of the deployment (Fig. 11a). $T_a$ followed a similar trend to $T_w$ rising from -7.6 to -4.1 °C. The net heat loss $Q_n$ steadily decreased from 165 W m$^{-2}$ to 12 W m$^{-2}$ (Fig. 11b) due to an increase in the magnitude of $Q_{sw}$. In Fig. 11c $\mu_f$ fluctuated

between 1 mm and 10 mm during the deployment with an average of 4 mm. In Figs. 11d-e the time series of number and volume concentrations did not exhibit significant trends. $C_{fn}$ ranged from $4.1 \times 10^{-5}$ cm$^{-3}$ to $2.4 \times 10^{-3}$ cm$^{-3}$ with an average of $5.6 \times 10^{-4}$ cm$^{-3}$ while $C_{fv}$ was negligible most of the time with occasional spikes up to $4.2 \times 10^{-4}$. One spike that occurred at 10:39 caused both $C_{fn}$ and $C_{fv}$ to reach their peak values. Visual examination of the images shows that at this time the number of flocs increased significantly for three consecutive images and this was possibly caused by a large floc colliding with the camera frame and fracturing.

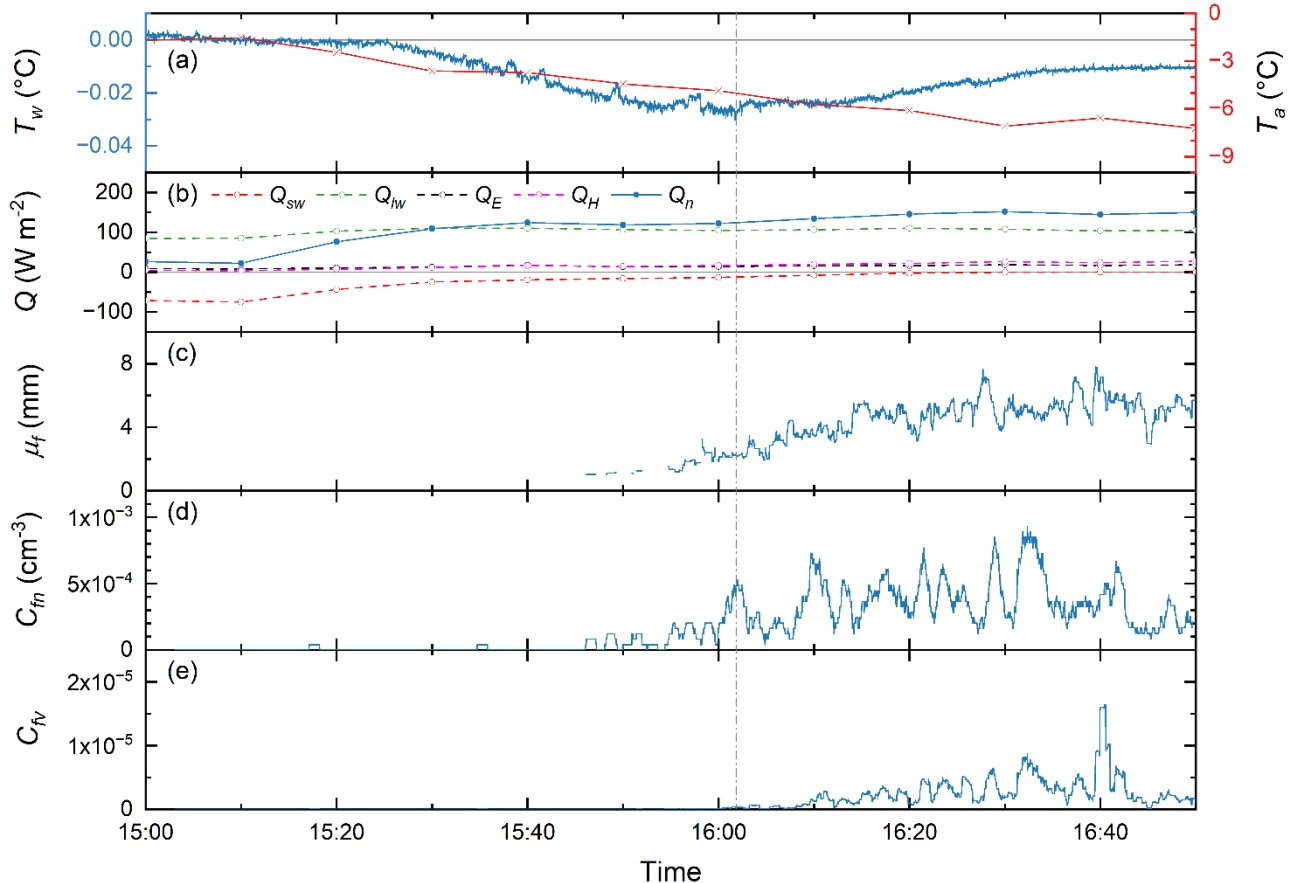

Figure 9. Time series of (a) water and air temperatures $T_w$ and $T_a$, (b) heat flux $Q$, (c) floc mean size $\mu_f$, (d) floc number concentration $C_{fn}$ and (e) floc volumetric concentration $C_{fv}$ for deployment NSR-L4 on December 12, 2021. The vertical dashed grey line indicates the time when the peak supercooling temperature is achieved.

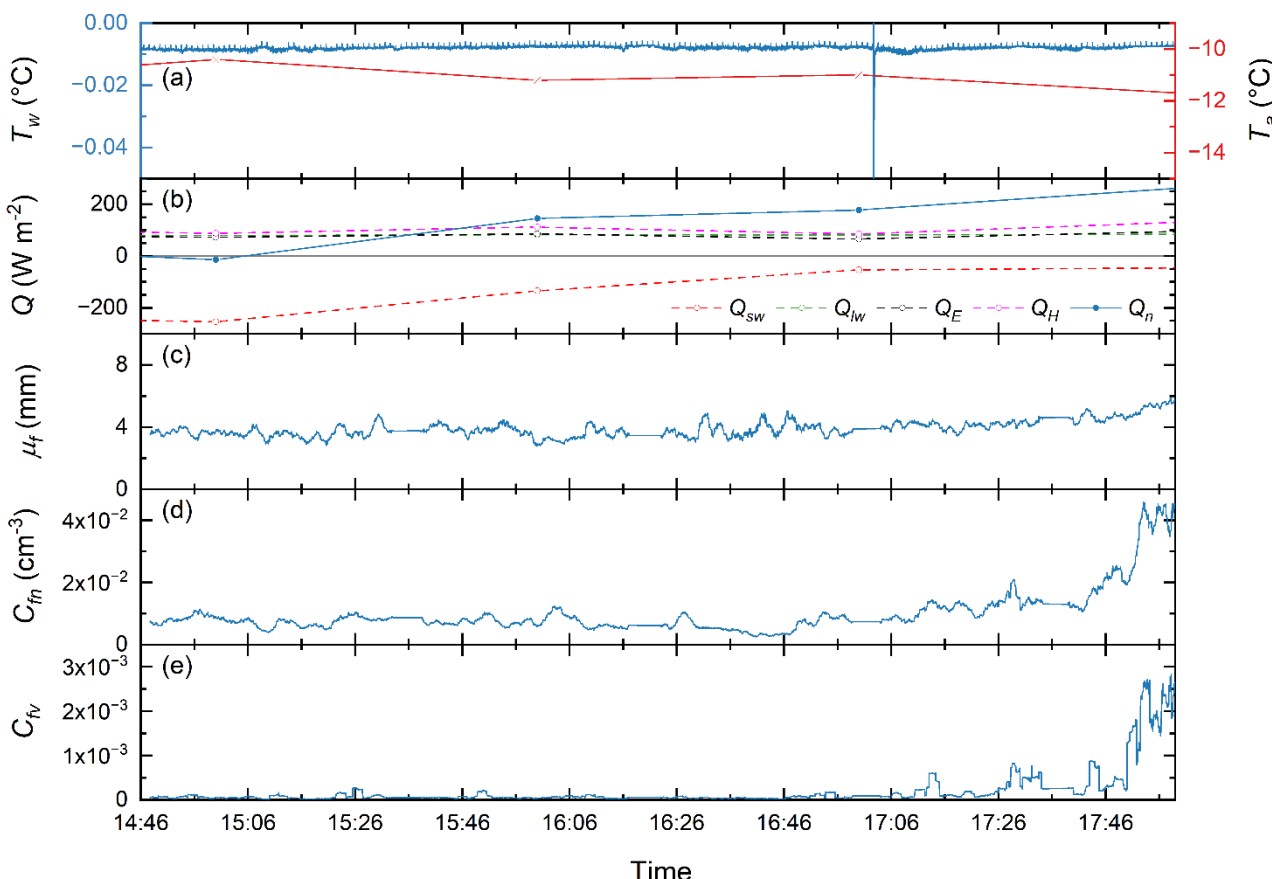

416

**Figure 10.** Time series of (a) water and air temperatures $T_w$ and $T_a$, (b) heat flux $Q$, (c) floc mean size $\mu_f$, (d) floc number concentration $C_{fn}$ and (e) floc volumetric concentration $C_{fv}$ for deployment KR-F1 on January 30, 2023.

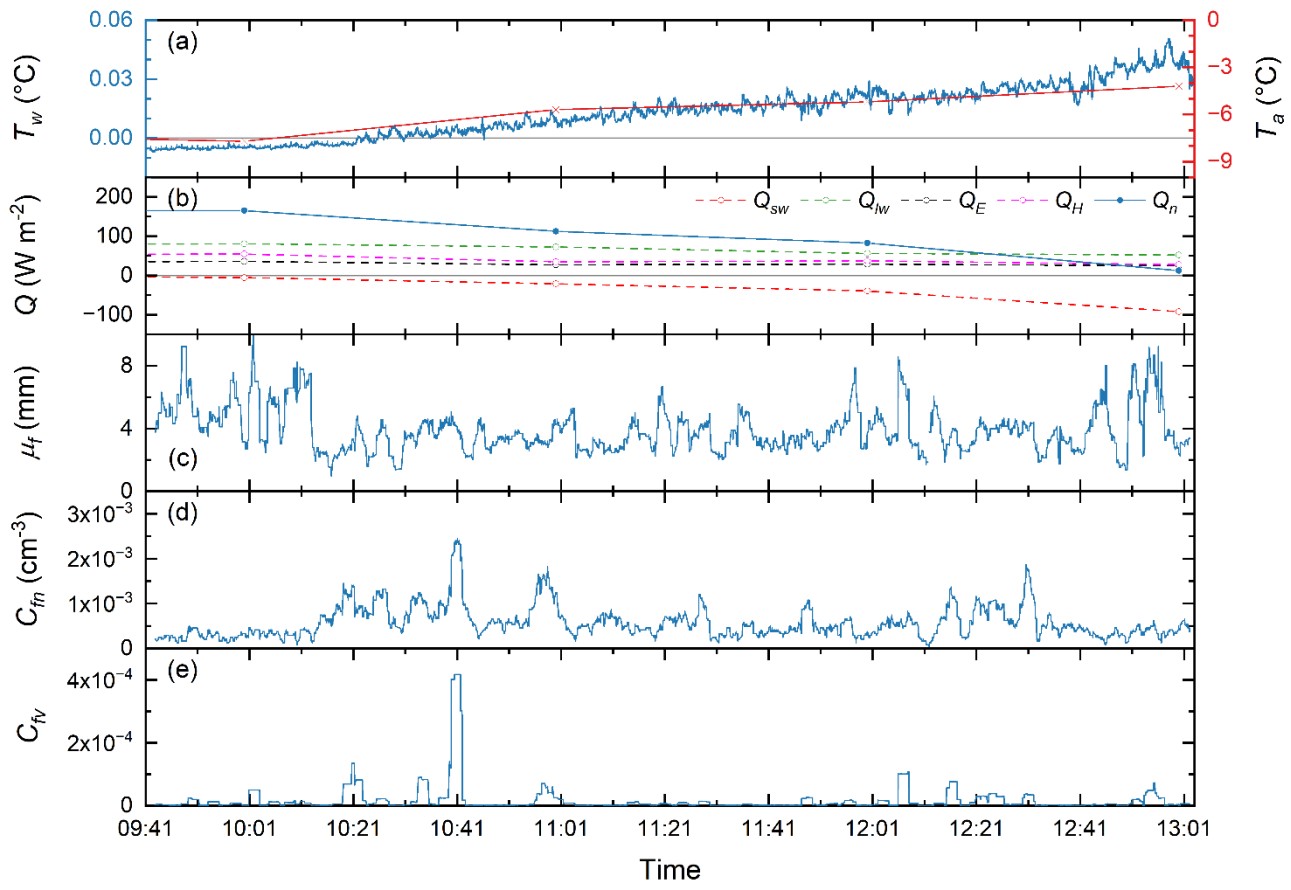

419

Figure 11. Time series of (a) water and air temperatures $T_w$ and $T_a$, (b) heat flux $Q$, (c) floc mean size $\mu_f$, (d) floc number concentration $C_{fn}$ and (e) floc volumetric concentration $C_{fv}$ for deployment PR-F2 on December 13, 2022.

## 6 Discussion

Images of typical frazil flocs shown in Fig. 7 illustrate the complexity of their morphology, which encompasses various ice crystal shapes, including disc-shaped, needle-shaped, and irregular particles. Disc-shaped ice particles were observed in flocs from all three rivers but were most pronounced in the NSR where flocs were almost all formed by disc-shaped particles of different sizes (Figs. 7a~b). The growth of frazil ice in supercooled water is limited by the diffusive removal of the latent heat of solidification from the ice-water interface and by the slow attachment kinetics in the perpendicular direction, which leads to the formation of disc-shaped particles (Mullins and Serkerka, 1964; Rees Jones and Wells, 2015). Flocs containing needle-shaped crystals as shown in Fig. 7c were observed during deployment PR-F1 which had a very low mean air temperature of -20.64 °C. These types of crystals have been found to form primarily at the surface of supercooled water (Hallett, 1959; Clark and Doering, 2002). The cold air temperature during deployment PR-F1 may have promoted the growth of these needle-shaped

particles at the water surface before they were entrained in the water column and subsequently attached to flocs. Irregular particles were observed in flocs from both the KR and PR, most pronouncedly in deployment KR-E1 as shown in Fig. 7d. Irregularly shaped particles are formed by unstable disk growth which is known to be caused by the formation of temperature gradients in the water surrounding the particles (Kallungal and Barduhn, 1977). This suggests that during the KR and PR deployments, frazil ice particles probably spent some time in relatively quiescent water where the turbulence intensity was so low that temperature gradients could form in the water surrounding the particles. Another possibility is that the particles were temporarily transported to the river surface exposing them to cold air, which may also lead to unstable disk growth. In addition, broken fragments of skim ice or border ice that were entrained into the water column are another possible source of irregular particles in flocs. Clark and Doering (2009) observed in the laboratory that flocs could become denser over time when the turbulence intensity was higher. During deployment KR-E1, although the locally measured depth-averaged velocity near the FrazilCam was relatively low at 0.22 m s$^{-1}$, the water velocity ~70 m upstream of the deployment site was visually observed to be very turbulent due to the presence of four groins and a narrow channel width. Therefore, this may have contributed to the denser flocs that were observed during this deployment.

The data presented in Table 4 and Fig. 8 are the first quantitative measurements of frazil floc sizes and concentrations in rivers. The mean floc size averaged over all deployments was 3.80 mm, which was close to the mean values observed for most of the individual deployments except for deployments NSR-L3, NSR-L6, and KR-E1 which had mean floc sizes of 1.87, 1.19, and 5.64 mm, respectively. As noted in Sect. 5.1, flocs observed during deployments NSR-L3 and NSR-L6 were comprised of much smaller disc-shaped individual particles (Fig. 7b) than the rest of the deployments (Fig. 7a). Deployment NSR-L3 took place during a principal supercooling event in which the observed small frazil ice particles were likely newly formed and still growing, which could be the reason why the flocs were smaller and comprised of significantly smaller particles. In addition, deployment NSR-L3 took place as the crest of a hydropeaking wave was passing the site that resulted in a mean water depth of 1.24 m which is 37% to 55% larger than the depths during the other NSR deployments (Table 3). The significantly higher water depth reduced the fractional height where the images were collected, which could also result in smaller floc sizes. This would be consistent with measurements by Reimnitz et al. (1993) that showed that larger flocs have higher rise velocities. Deployment NSR-L6 occurred during the 2022 freeze-up season, which was the shortest freeze-up in ~10 years lasting only three days. Significantly smaller flocs were observed during this deployment (see Fig. 7b) and this may be because smaller relatively young flocs were generated during this rapid freeze-up process. The largest mean floc size, maximum floc size and largest concentration (see Table 4) were observed during deployment KR-E1 (Fig. 7d). As discussed previously the particles that formed flocs during KR-E1 included irregularly shaped particles and this could have resulted in larger flocs compared to flocs formed entirely by disc-shaped particles.

The mean floc size and standard deviation ranged from 1.19 to 5.64 mm, and 0.88 to 5.03 mm, respectively as shown in Table 4. The 95$^{th}$ percentile of floc size ranged from 2.47 to 14.28 mm, and the largest flocs found was 99.69 mm in size. Schneck

et al. (2019) conducted laboratory experiments in a frazil ice tank with an average turbulent dissipation rate of 0.034 $m^2$ $s^{-3}$
which falls within the range of the values estimated in the three rivers in this study (0.005 ~ 0.207 $m^2$ $s^{-3}$). They found that in
freshwater the size distribution of flocs followed a lognormal distribution and the mean size, 95[th] percentile of floc size, and
maximum size were 2.57 mm, 6.91 mm, and 95.1 mm, respectively. The mean and 95[th] percentile sizes fall within the range
of the values observed in this study. However, the overall mean floc size observed in the field was 3.80 mm, which is 48%
larger than the mean measured in the laboratory. The maximum floc sizes observed in the laboratory and field are comparable.
It is worth noting that the largest floc size of 99.69 mm was just slightly smaller than the FOV dimensions and considerably
larger than the 3.6 cm gap, indicating that the floc size measurements may have been physically limited by the FOV and the
gap between the polarizers. Therefore, further increases in both the FOV and the gap between the polarizers may be needed in
future studies to allow even larger flocs to be imaged and measured.

The size distributions obtained from different rivers are all a reasonable visual fit to a lognormal distribution as shown in Fig.
8, which is consistent with the laboratory measurements (Schneck et al., 2019). However, when the Chi-square test for
goodness-of-fit was applied none of the size distributions were quantitatively confirmed to fit a lognormal distribution at the
5% significance level.  This could be primarily due to the use of the cut-off size to eliminate sediment particles which produced
a sharp cut-off in the distributions. In addition, the small number of samples in some deployments resulted in noisy size
distributions making it unlikely that they would be a good quantitative fit to a smooth lognormal distribution. Nonetheless, the
good qualitative comparison of the floc size distributions measured in the field with theoretical lognormal distributions in Fig.
8 does suggest that if the sample size was larger and sediment particles could be filtered out that floc size distributions in rivers
would also closely follow a lognormal distribution.

The average floc number concentration $\overline{C_{fn}}$ ranged from $1.80 \times 10^{-4}$ to $1.15 \times 10^{-1}$ $cm^{-3}$ (Table 4), Schneck et al. (2019)
measured a peak floc number concentration of $2.5 \times 10^{-1}$ $cm^{-3}$ in freshwater laboratory experiments, which is similar in
magnitude to the upper limit of values measured in the field. The average floc volumetric concentration $\overline{C_{fv}}$ ranged from 2.05
$\times 10^{-7}$ to $4.56 \times 10^{-3}$ (Table 4). Previous studies reported suspended ice volumetric concentrations ranged from $2 \times 10^{-6}$ to $6 \times$
$10^{-3}$ (Tsang, 1984; Marko and Jasek, 2010; Richard et al., 2011). These measurements were made using comparative resistance
probes and acoustic devices which in theory detect all of the ice suspended in the water. The upper range of previous
concentration measurements is comparable to that reported in this study. However, the lower range is one order of magnitude
larger than this study, which may be due to the fact that the previous studies reported the total volume of frazil flocs and
particles.

The time series of frazil floc properties in Fig. 9 indicate that during the principal supercooling phase, floc number and mean
size started to increase significantly just prior to peak supercooling and reached a maximum near the end of principal
supercooling, the floc volumetric concentration only started to increase significantly after peak supercooling occurred.
Deployment NSR-L3 that captured almost the entire principal supercooling phase also showed a similar trend (see Fig. S3 in
the Supplement). The increasing trend of floc mean size and number concentration generally agrees with previous laboratory
measurements (Schneck et al., 2019; Pei et al., 2023). However, laboratory measured mean floc size and number concentration
stopped increasing significantly shortly after peak supercooling, while in the field they stopped increasing later, near the end
of the principal supercooling period. For example, Schneck et al. (2019) observed that the mean floc size and number
concentration in freshwater stopped increasing significantly at dimensionless times of $t / t_c = 1.13$ and $1.27$, respectively
compared to $t / t_c = 2.00$ and $1.81$ for NSR-L4 ($t_c$ is the time interval between the start of supercooling and peak supercooling
and $t$ is the time). The peak floc number concentration measured during the three Principal deployments in this study ranged
from $9.3 \times 10^{-4}\,cm^{-3}$ to $3.1 \times 10^{-3}\,cm^{-3}$, which was more than two orders of magnitude lower than the $2.5 \times 10^{-1}\,cm^{-3}$ measured
in the laboratory tank by Schneck et al. (2019). These significantly lower floc concentrations suggest that particle
concentrations in the field were also much lower compared to laboratory measurements. At lower suspended frazil
concentrations the collision frequency of frazil particles would be reduced, increasing the time for flocs to gain mass via
collision-induced particle-floc aggregation, which might explain the longer time period that mean floc size and number
concentration was observed to increase in the field.

Figure 10 shows that during KR-F1 the mean floc size was approximately constant prior to the arrival of the hydropeaking
wave during the residual supercooling phase. Similarly, there were no trends observed in floc size in five other Residual
deployments, NSR-L2, NSR-L5, KR-E1, PR-F1(see Figs. S2, S4, S7 and S6 in the Supplement) and PR-F2 (Fig. 11).
McFarlane et al. (2019b) found that in rivers the mean particle size remained approximately constant during the residual
supercooling phase if the environmental conditions were relatively stable. Therefore, it follows that flocs observed during the
residual supercooling phase would also have a stable mean size unless hydraulic and/or meteorological conditions changed
significantly. The mean floc size is the most stable during deployment KR-E1 (Fig. S7 in the Supplement) with a fluctuation
range of only 1.5 mm, which could be in part due to the significantly larger sample size of 187,288. The only two Residual
deployments that did not have a stable mean floc size were NSR-L6 and KR-F2 (Figs. S5 and S8 in the Supplement), and in
both cases, the size decreased and this coincided with minor increases in $T_w$ (~0.005 °C). These results indicate that during the
residual phase the mean floc size does not typically vary significantly even at the end of the supercooling event when $T_w$ rises
above zero, as was the case in PR-F1 and PR-F2. During the two PR deployments the floc properties did not change
significantly during the 1.3- and 2.5-hour time periods between when supercooling ended, and the measurements stopped. This
is likely because the zero degree isotherm had moved upstream of the deployment site but the frazil being generated upstream
of it was still advecting past the FrazilCam (i.e., the zero degree isotherm was not so far upstream that the advecting frazil had
time to melt.)

As shown in Fig. 10, during KR-F1 the residual supercooling water temperature remained mostly approximately constant at a
temperature of approximately -0.01°C. An approximately constant residual supercooling temperature was also observed in
NSR-L2, KR-E1 and NSR-L5 (see Figs. S2, S7, and S4 in the Supplement). This means that during the residual supercooling
phase ice was still growing and releasing latent heat that balanced the heat loss from the water surface in order to maintain the
approximately constant water temperature. In this study, although the mean floc size did not vary significantly during most of
the measured residual supercooling deployments, fluctuations and trends in the floc number and volume concentration time
series were observed. This indicates that there may have been frazil ice particles still forming and growing, releasing latent
heat to help balance the surface heat loss. In addition, during the residual phase anchor ice, border ice, and surface ice pans
were likely growing as well and releasing latent heat, helping to maintain the stable residual supercooling temperature.

The time series of water temperature $T_w$ and net heat flux $Q_n$ provided an opportunity to theoretically estimate the total ice
growth in the water column, which could be compared to the measured floc volumetric concentration $C_{fv}$ to estimate the
fraction of ice sampled by the FrazilCam. Assuming there were no significant water temperature gradients in any direction
(i.e. the river had a uniform temperature) and that the water depth was constant, the thermal balance of the water-ice mixture
is given by:
$\rho C_p \frac{dT_w}{dt} = -\frac{Q_n}{\bar{d}} + \rho_i L_i \frac{dC_i}{dt}$,                                    (15)
where $\rho$ is the water density, $C_p$ is the specific heat of water, $\rho_i$ is the ice density, $L_i$ is the latent heat of fusion of ice, and $C_i$
is the total ice concentration due to thermal growth (Souillé et al., 2023). Eq. (15) was then used to estimate, $C_i$ for deployment
NSR-L4, which captured the entire principal supercooling period. The result showed that the FrazilCam was only sampling at
most 2% of the total ice that was forming in the water. It should be noted that $Q_n$ used in the calculation does not account for
the effect of surface ice due to a lack of accurate surface ice data. In addition, mean water depth $\bar{d}$ was used while in reality
water depth varied spatially and temporally. These approximations create considerable uncertainty in the calculations of the
total heat loss from the surface, and the volume of the water being cooled. Given all the simplifying assumptions that were
made the uncertainty in the calculated $C_i$ is potentially quite large, but is likely not greater than a factor of two or three.
Therefore, despite this potential large uncertainty, the calculations suggest that the FrazilCam was only sampling less than
~5% of the total ice being formed in the river. Similar calculations were also performed using data collected in a laboratory
frazil ice tank experiment using the laboratory version of the FrazilCam. In the laboratory environment the water depth is a
constant, the tank has been shown to be well mixed and the surface heat loss can be quantified from the water cooling rate
with reasonable accuracy. These results showed that $C_i$ calculated using Eq. (15) was comparable to the volumetric
concentration of suspended ice calculated from the FrazilCam images prior to when flocs began rising to the surface. This
demonstrates that the FrazilCam does provide accurate measurements of the suspended ice concentrations. However, the only
time the FrazilCam would be sampling a significant fraction of the total ice being formed in a river would be when suspended
frazil is the only ice that is actively growing.

The effect of surface heat flux on floc properties was investigated. A positive mean net heat flux $\overline{Q_n}$ was observed for all
deployments indicating a net heat loss from the surface. The magnitude of $\overline{Q_n}$ ranged from 95.4 to 318.8 W m$^{-2}$ as shown in
Table 4. The dominant positive heat flux was $Q_{lw}$ and $Q_H$ for six and five deployments, respectively, while the dominant
negative heat flux in all deployments was $Q_{sw}$ which is consistent with previous studies (McFarlane and Clark, 2021; Boyd et
al., 2023). Efforts were made to correlate the mean net heat flux $\overline{Q_n}$ with the measured floc properties listed in Table 4 (i.e.,
columns 5~11). No significant correlations were found when using data from all deployments or when only the data from the
six NSR deployments that have 10-min heat flux data were used. It is worth noting that the heat flux analysis in this study did
not account for varying surface ice concentrations and neglected several heat fluxes (e.g. sediment-water). Clearly, more
comprehensive and frequent measurements of heat fluxes and surface ice properties are needed in future studies to more fully
understand the impact of varying heat fluxes on frazil floc properties.

To investigate the effect of hydraulic conditions on the mean floc size $\mu_f$, the local Reynolds number $Re$ is plotted versus $\overline{\mu_f}$
in Fig. 12 along with the following linear regression equation:
$$\overline{\mu_f} = 6.82 - 3.05 \times 10^{-5} Re ,  \qquad (16)$$
As $Re$ increases from ~40,000 to ~160,000, $\overline{\mu_f}$ decreases from approximately 5.5 mm to 2 mm and the coefficient of
determination ($R^2$) is 0.69, indicating that the two are moderately correlated. Clark and Doering (2009) found that higher
turbulence intensity inhibited the formation of large flocs. This finding is consistent with the correlation presented in Fig. 12
if it is assumed that turbulence intensity increased with $Re$ in the three study rivers. However, this is not necessarily the case.
An alternate explanation for the observed correlation is that as $Re$ increased flocs experienced higher shear strain rates (i.e.,
larger velocity gradients) and more violent floc-floc collisions which would tend to rupture larger flocs and reduce their mean
size.

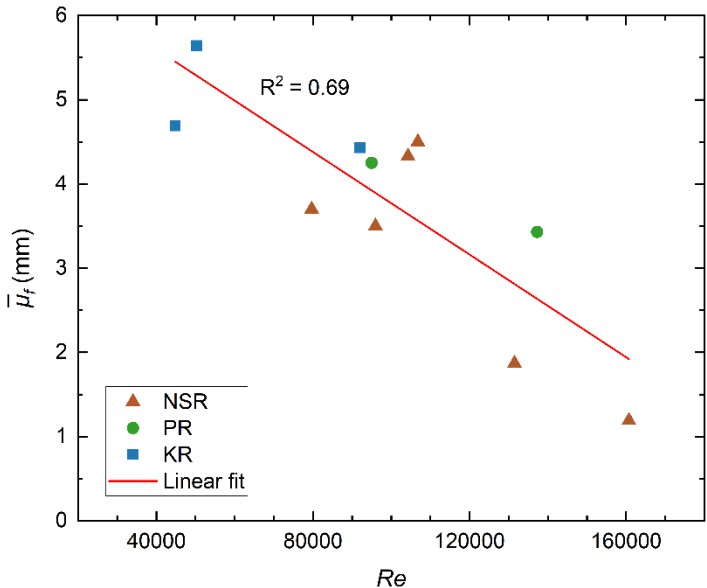


**Figure 12. Relationship between local Reynolds number _Re_ and mean floc size $\overline{\mu_f}$ in mm.**


The effect of water depth on the floc volumetric concentration was investigated by correlating the average volumetric
concentration with the fractional height $d_m/\bar{d}$ where $d_m = 0.198\,m$ is the height above the bed at the centre of FrazilCam
FOV and $\bar{d}$ is the mean water depth. Figure 13 presents a scatter plot of the fractional height $d_m/\bar{d}$ versus the average floc
volumetric concentration $\overline{C_{fv}}$. Results show that there is a strong nonlinear correlation given by the following power law
equation:
$$\overline{C_{fv}} = 4.80 \left(\frac{d_m}{\bar{d}}\right)^{9.69},\qquad\qquad\qquad\qquad\qquad\qquad\qquad\qquad\qquad\qquad(17)$$
where the $R^2$ value equals 0.99. Ye (2002) and Morse and Richard (2009) reported measurements of vertical frazil
concentration profiles and found that the Rouse equation (Rouse, 1937), previously used to characterize suspended sediment
concentration profiles, could be used to describe the frazil ice concentration profile. Equation (17) is similar in format to the
Rouse equation, indicating that the vertical concentration of both frazil particles and flocs may be accurately described by
power law equations.

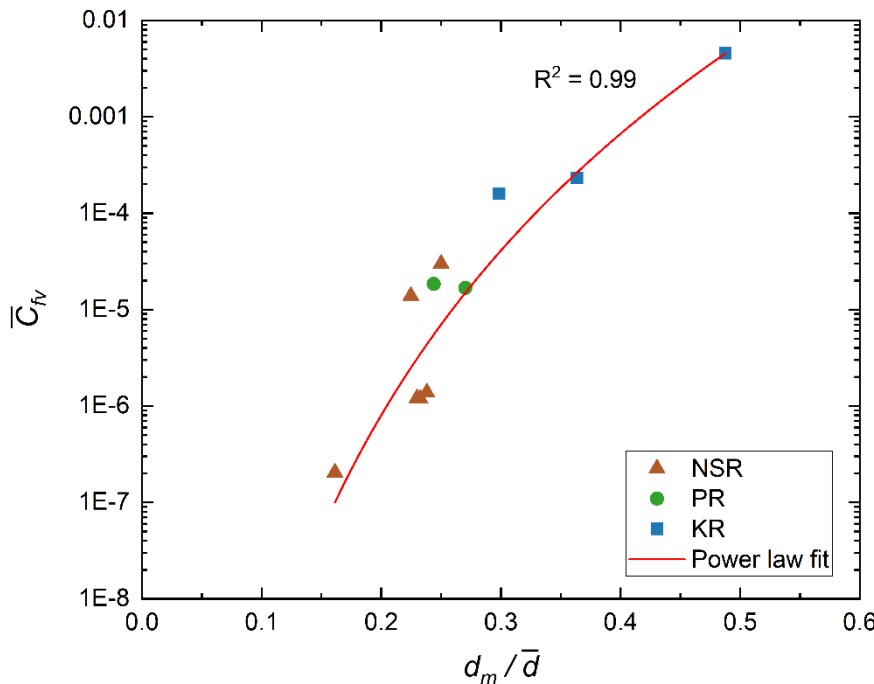


**Figure 13. Relationship between the fractional height $d_m/\bar{d}$ and the average floc volumetric concentration $\overline{C_{fv}}$**
**7 Conclusions**
A submersible high-resolution camera system was deployed during supercooling in three rivers from 2021 ~ 2023. Images
from the eleven deployments were analyzed to investigate frazil floc properties and their evolution. Images showed that frazil
flocs observed in the North Saskatchewan River were predominately formed by disc-shaped particles, while flocs in the Peace
River and Kananaskis River were comprised of various ice crystal shapes, including disc-shaped, needle-shaped, and irregular
particles. A lognormal distribution is a reasonable description of floc size distributions in rivers. The mean floc size ranged
from 1.19 to 5.64 mm and the overall mean floc size was 3.80 mm. The mean floc size in rivers was found to 48% larger than
was previously observed in the laboratory by Schneck et al. (2019) while the maximum floc size was comparable in the
laboratory and field. The average floc number concentration ranged from $1.80 \times 10^{-4}$ to $1.15 \times 10^{-1}$ cm$^{-3}$ and previous laboratory
measurements fall within the range of the values observed in this study. The estimated average floc volumetric concentration
ranged from $2.05 \times 10^{-7}$ to $4.56 \times 10^{-3}$, with the upper bound being comparable to previous total ice volume concentration
measurements while the lower bound is an order of magnitude smaller.

Time series analysis indicated that during the principal supercooling phase, floc number concentration and mean size increased
significantly just before peak supercooling and reached a maximum near the end of principal supercooling. This increasing
trend was also observed in previous laboratory measurements (Schneck et al., 2019; Pei et al., 2023) but the duration of the
increasing trend was longer in the field. During the residual supercooling phase, the mean floc size did not typically vary
significantly even 2.5 hours after the water temperature rises above zero degrees. The effect of the air-water heat flux on floc
properties was investigated by conducting a linear regression analysis. However, no significant correlations were found, and
this may be due to the limited dataset or the complexity of the field environment where heat fluxes can vary temporally and
spatially. Future field measurements of floc properties, especially made during the principal supercooling phase and made
continuously along multiple sites along a study reach, are needed to more fully understand the factors that govern their size
and concentration.

Analysis of the influence of local hydraulic conditions on frazil floc properties showed that as the local Reynolds number
increases, the mean floc size decreases linearly. The resulting equation can be used to estimate mean floc sizes in rivers using
estimates of the mean velocity and depth. It was also shown that the averaged floc volumetric concentration can be related to
the fractional height above the bed through a power law equation. This relationship may be useful for describing the vertical
concentration profiles of frazil flocs.

The detailed measurements of frazil floc properties and their evolution in rivers presented in this study could be used in several
ways to enhance numerical modelling of river ice processes in order to improve predictions of river freeze-up. At the present
time the frazil rise velocity is treated as a calibration parameter in comprehensive river ice process models (e.g. Shen, 2010;
Blackburn and She, 2019). However, it could now be directly estimated by first using Eq. (16) to predict the mean floc size
using the local Reynolds number and then the rise velocity could be predicted using Reimnitz et al. (1993) measurements. In
addition, the reported lognormal size distributions of flocs, as well as time series evolution of mean floc size and
concentrations, measured in rivers for the first time, could provide opportunities to incorporate floc dynamics into numerical
models with the goal of improving how realistically they simulate frazil ice evolution and surface ice progression.

In the future, it would be of interest to deploy the FrazilCam in lakes and oceans, where the flow regime and salinity may be
considerably different, to investigate frazil particle and floc properties in these different environments. The FrazilCam system
in principle can be deployed in any sufficiently transparent waters, however, the system would need to be modified to automate
the polarizer rinsing process. This would be challenging but might be possible using a mechanical wiper which would allow
deployments on the bottom of deeper water bodies. In addition, the system could be attached to an unmanned or autonomous
underwater vehicle to allow observations to be made at various depths in the water column in lakes and oceans.

## Code and data availability

Part of the meteorological data used to carry out heat flux analysis were obtained from Alberta Climate Information Service (ACIS) http://agriculture.alberta.ca/acis/weather-data-viewer.jsp, Environmental and Climate Change Canada (ECCC) https://climate.weather.gc.ca/historical_data/search_historic_data_e.html, and University of Calgary Biogeoscience Institute https://research.ucalgary.ca/biogeoscience-institute/research/environmental-data. Historic sediment data for the North Saskatchewan River at Edmonton and Peace River at Dunvegan Bridge can be accessed from Water Survey of Canada Historical Hydrometric Data https://wateroffice.ec.gc.ca/mainmenu/historical_data_index_e.html. All other data and code used in this study are available from the authors upon request.

## Author contribution

CP and JY prepared the apparatus and performed the field work together with advice from YS and ML. CP carried out the analysis and processing of the data, prepared the figures, and wrote the manuscript with review and contributions from JY, YS, and ML.

## Competing interests

The authors declare that they have no conflict of interest.

## Acknowledgements

This research was supported by the Natural Sciences and Engineering Research Council of Canada (NSERC) (Grant nos. RGPIN-2021-02887, RGPAS-2021-00022 and RGPIN 2020-04358). We would like to thank Heyu Fang, Xun Hong, and Vincent McFarlane for their assistance in field deployments. We would also like to thank Perry Fedun for providing technical assistance. The first and second authors are partially funded by the China Scholarship Council (CSC) and the University of Alberta, respectively. Both are gratefully acknowledged. The authors thank an anonymous reviewer and Dr. Steve Daly for their insightful comments that helped to improve the manuscript.

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
