# Peer review of "Measurements of frazil ice flocs in rivers"

_EGUsphere, 2024_

## Author Comment (AC1)

**Authors responses to comments posted by Referee #2**

General comments and suggestions:

The authors present a well written study which investigates the properties and concentrations of frazil flocs in reaches of the North Saskatchewan River (NSR), Peace River (PR), and Kananaskis Rivers (KR) using a submersible camera deployment called FrazilCam. The manuscript presents in-situ observations of floc shape, size, and concentration which is of use to the frazil and river ice communities for improved modelling efforts. The following general suggestions can be considered by the authors for potentially improving the quality of the manuscript. Specific comments follow these general suggestions.

> The authors thank Referee #2 for the constructive comments and helpful suggestions. We are confident that the recommended revisions will improve the quality of the manuscript. Please find our responses below in blue.

1. The methodology used for classification of flocs was challenging to follow at times. Perhaps adding in a flowchart or process-flow diagram would aid in reader comprehension to better follow all the processing steps.

> We agree and will add a flowchart to the revised manuscript.

2. The heat flux analysis requires additional clarification and minor reanalysis. Namely, description of equations used and some minor addendums to methodology. Please see specific comments.

> We will revise Sec. 4.2 Heat flux analysis to include a description of equations used and justifications for why we chose these equations.

Detailed comments and suggestions:

Study Reaches:

L108-111: Is the data obtained from Kellerhals et al. (1972) the most up to date site data available?

> Yes. To our knowledge it is up to date and is representative of the current characteristics of the study reaches.

L114: The site map is well made and clear.

> Thanks!

Instrumentation, Methodology, and Deployments:

L115 and L126: The study modifies the FrazilCam system developed by MacFarlane et al. (2017). Can the specific modifications from the original system be made more clear in this section? Was the FOV the only aspect modified (L126-127).

The system was modified to enlarge the FOV and increase the gap between the polarizers. These sentences will be revised to read:

"The camera system was modified to enlarge the FOV and increase the gap between the polarizers to enable larger flocs to pass through and fit within the FOV. The modified configuration has an ~6 times bigger FOV and 1.6 times larger gap compared to McFarlane et al. (2017)"

L147: Which frequency Aquadropp ADCP was used? My concern is for blanking distances, as the river sites have quite shallow depths (Table 3).

We used a 2 MHz AquaDopp High Resolution Profiler with a blanking distance of 0.1 m. As mentioned in L146, we only used the velocities measured at 60% of the water depth to get estimates of depth-averaged water velocity so the blanking distance should not be an issue. We will revise this paragraph to make it clearer.

Image Processing:

L221-223: Were the preliminary experiments conducted by yourself? If not, please provide some more context.

Yes, the preliminary experiments were conducted by myself.

L234: 'S' is not defined prior to its use here.

*S* was previously defined at L205.

L237: Perhaps it should be made clearer earlier in the study that only 1 deployment coincided with an entire 'principal' supercooling event.

We agree and will add a sentence in Sec. 3 right before Table 3 to make this clear earlier.

L231-264: The addition of a process-flow diagram or flowchart would largely help the reader understand the methodology used. Additionally, it would be useful to consider provided quantitative measurements of how many images were taken (in total), followed by how many were removed at each processing step.

> The requested diagram will be added and Table 3 will be revised to add number of images processed for each deployment as suggested. We will also add that images removed during the rinsing period constituted 2~14% of the total number of images.

L255: One key missing piece of information was the specific sampling time used for each deployment. On L131, it is noted that 5 images at 1 Hz every 9,15, or 19s were acquired depending on field conditions. It would be more transparent to describe under what case/deployment each sampling time was used. If sampling times were mixed for a given deployment, this should be also stated and justified.

> We agree and will revise Section 3 right before Table 3 to provide the following information: NSR-L1 to L6 and KR-E1 used 5 images every 9s; PR-F1 and F2 used 5 images every 18s; and KR-F1 and F2 used 5 images every 15s, sampling times were not mixed for any given single deployment. Note that, we noticed a typo in the manuscript - the third time interval is 5 images at 1 Hz every 18s, not 19s and this error will be corrected in the revised manuscript.

Heat Flux Analysis:

L273: It is uncommon in heat flux analysis to explicitly consider an albedo in the longwave spectrum as nearly all radiation in this spectrum is thought to be absorbed at/near the surface (shown by your use of a very small albedo of 0.03). It is recommended to remove this.

> The albedo of 0.03 in the longwave radiation calculation was calculated for water temperatures of normal ranges (Raphael, 1962). It has been used in multiple recent river energy budget studies in the river ice fields (e.g., Richard et al., 2015; McFarlane and Clark, 2021; Yang et al. 2023a). Therefore, we decided to keep the albedo of 0.03 to be consistent with these studies.

L277: Mean water temperatures are used here for conducting a surface energy balance. While vertical turbulence may be well developed and river depths relatively shallow, some degree of caution should be presented on this matter. Surface temperatures may deviate significantly depending on flow and meteorological conditions. The assumption that the river reach is fully vertically mixed should be stated explicitly in this case.

> We agree and will state this assumption explicitly as recommended.

L278-280: Satterlund (1979)'s parameterization relies on data from Aise and Idso (1978) from continental Montanna and is extended with Stoll and Hardy (1955) for measurements in Alaska. Perhaps it may be more prudent to select a more well-used scheme for clear sky conditions shown effective in higher latitude regions of North America (e.g. Efimova, 1961). Key et al. 1996 provide a review on the matter using data from Alaska and the Northwest Territories on their

review of parameterization schemes. It is left to the authors' discretions to keep the current scheme or adopt one of the above-mentioned after reviewing the noted references.

Efimova, N. A. (1961). On methods of calculating monthly values of net longwave radiation. Meteorol. Gidrol., 10, 28-33

Key, J.R., Silcox, R.A., Stone, R.S., 1996. Evaluation of surface radiative flux parameterizations for use in sea-ice model. J.Geophys. Res. 101 C2, 3839–3849

Thank you for this suggestion. In a previous study (Yang et al., 2023b), the co-authors investigated various formulas used to calculate downwelling longwave radiation and the latent and sensible heat fluxes during freeze-up on the North Saskatchewan River. Formulas for the downwelling longwave radiation for clear-sky atmospheric emissivity and cloud effect were assessed using direct measurements. The combination of formulas used in this study were the ones that provided the most accurate results in Yang et al (2023b). Therefore, we decided to keep the current scheme.

L281: Where was Bowen's ratio obtained from? In addition to the above-mentioned, please describe equations used for the flux analysis within this section. It provides the reader with the information readily, rather than having to access several other sources to understand the approach taken.

The Bowen's ratio is given by:

$$B = \frac{C_a P}{0.622 l_v} \times \frac{T_s - T_a}{e_s - e_a},$$

where $C_a$ is the specific heat of air; $l_v$ is the latent heat of vaporization; $P$ is the atmosphere pressure; $T_s$ is the surface water temperature taken the same as the measured water temperature. $T_a$ is the air temperature; $e_s$ and $e_a$ are the saturated and actual vapour pressure of water, respectively.

We will revise this section as suggested to include all the equations used for the heat flux analysis as well as justifications for why we chose these equations.

Results:

Floc Shape, Size and Concentration:

L303: It was quite interesting to record such a large floc size (99.69mm). As I understood, the FrazilCam in this study had an increased FOV relative to its predecessor. Would recording a max floc size such as this in KR-E1 suggest perhaps the FOV may need to be further increased? Perhaps there may be potential for biasing floc sizes too low, as larger flocs that are unfavourably oriented interact with the polarizers and break.

Yes, this does suggest that further increases in both polarizer size and the gap between the polarizers might be needed to allow even larger flocs to be imaged. We will add a short discussion to the revised manuscript regarding the possibility that the current system may be under-sampling the large flocs.

Floc Size Distribution:

L319: You note that lognormal distributions are reasonable fits for the distributions. Would you be able to provide a quantitative measure of the fit for each of the histograms?

This statement was based only on visual examination that a theoretical lognormal distribution was a reasonable fit to the observed distributions. As suggested we attempted to provide quantitative evidence for this statement. We applied the Chi-Squared test and none of the distributions passed the test with a 5% significance level. This could be due to various reasons. First, as discussed in Sec. 5.2, the use of cut-off size to eliminate sediment particles produces a sharp cut-off in the distributions, the effect is more pronounced in deployments that recorded smaller floc sizes. Second, the small number of samples in some deployments resulted in noisy size distributions making it unlikely that they would be a good quantitative fit for a smooth lognormal distribution. We will revise the discussion section to make clear that our conclusion that the measured distributions are approximately lognormal is based on visual comparisons and that the Chi-Squared test did not statistically confirm the lognormal distribution.

Time Series:

L332: It is understandable that not all sites are presented within the contents of the manuscript. I do however believe that the reader is left curious as what the other sites might have looked like. Perhaps it can be considered to add in the other deployments data (similar or simpler versions to Figure 8,9,10) as supplementary data for further transparency.

This is a good suggestion and we will provide the other deployment data as supplementary data as recommended.

L357-358: The usage of hourly component flux data for correlating floc properties and concentrations is a bit questionable given the timescales of the deployments (~1-3hr). It is described between L155-167 where this data is obtained from. If these sites (minus the NSR reach) have sub-hourly data that can be used for the heat flux analysis, please consider updating.

Unfortunately, we do not have sub-hourly data for the KR and PR sites to update our results. Please note that the hourly heat fluxes were plotted in the time series only to illustrate the trends of the heat flux variations and to identify the dominant component during the deployments. We think this is useful because it facilitates better understanding of supercooling which is directly related to the surface heat flux. We did not try to correlate the hourly heat flux time series data with the floc properties or concentration time series data.

L375, 380, 383: Figures 8,9, and 10 would benefit from the addition of air temperature data.

We agree and will add the air temperature data as recommended.

Discussion:

L483-484: A larger limitation would be the use of hourly data rather than neglecting surface conditions and sediment heat fluxes. This is likely too coarse for the intended goal with correlating net surface heat flux with floc properties. This can be considered going forward for future studies on the matter.

We agree that this is a limitation of the study and we hope in the future to deploy our own weather stations to obtain meteorological data at faster sampling rates at these more remote sites. We did conduct a correlation analysis using only the heat flux data with 10-min resolution from the six NSR deployments and still did not find any significant correlations. Therefore, including the deployments with the 1-hour heat flux data did not change the current result. We will describe this analysis using only the 10-min data and explain that there was still no correlations found in the revised manuscript.

References

McFarlane, V. and Clark, S.P.: A detailed energy budget analysis of river supercooling and the importance of accurately quantifying net radiation to predict ice formation, Hydrol. Processes, 35(3), e14056, https://doi.org/10.1002/hyp.14056, 2021.

McFarlane, V., Loewen, M. and Hicks, F.: Measurements of the size distribution of frazil ice particles in three Alberta rivers, Cold Reg. Sci. Technol., 142, 100–117, https://doi.org/10.1016/j.coldregions.2017.08.001, 2017.

Raphael, J.M.: Prediction of temperature in rivers and reservoirs, J. Power Div. 88, 157–181, https://doi.org/10.1061/JPWEAM.0000338, 1962.

Richard, M., Morse, B. and Daly, SF.: Modeling frazil ice growth in the St. Lawrence River, Can. J. Civ. Eng., 42 (9), 592–608, https://doi.org/10.1139/cjce-2014-0082, 2015.

Yang, J., She, Y. and Loewen, M.: Assessing heat flux formulas used in the full energy budget model for rivers during freeze-up, in: CGU-HS Committee on River Ice Processes and the Environment (CRIPE) Proceedings of the 22nd Workshop on the Hydraulics of Ice Covered Rivers. Canmore, Canada. 9-12 July 2023, 2023b.

Yang, J., She, Y. and Loewen, M.: Assessing the uncertainties in modeling water temperatures during river cooling and freeze-up periods, , Cold Reg. Sci. Technol., 210, 103840, https://doi.org/10.1016/j.coldregions.2023.103840, 2023a.

---

## Author Response (AR1)

The authors wish to thank the two reviewers and Editor for the time and effort dedicated to providing feedback on the manuscript. We are grateful for the insightful comments that helped to improve the quality of the manuscript. We have made revisions as suggested and provided point-to-point responses to address all comments by the reviewers and Editor as presented below in blue. The line numbers in the responses refer to the tracked change manuscript.

**Response to Dr. Steve Daly**

This article is a valuable contribution to the literature on frazil ice formation in rivers. It is a well-written description of field observations made under difficult conditions using novel instrumentation developed by the authors. The article could very nearly be published as is. However, I do have very few specific comments. I also have some suggestions for the authors. I believe these suggestions would improve the paper, but it is not required that the authors make any of the suggested changes.

Line 14. The term "relative depth" confusing. Is this the distance from the bottom of the channel or the distance from the water surface?

> The relative depth is the ratio between the height from the river bed to the center of the field-of-view and the entire water depth. We think a clearer term is "fractional height" and used this in the revised manuscript. This revised the sentence in the abstract to (see L13-14):
>
> "The average floc volumetric concentration ranged from $2.05 \times 10^{-7}$ to $4.56 \times 10^{-3}$ and was found to correlate strongly with the fractional height above the river bed."

Line 22. It is suggested that the authors consider not using the term "sintering." There is a long history of using the term sintering with regard to ice. The very first uses were applied to the adhesion of ice particles in air when they were held together with some pressure. The reduction in surface energy of the system provides the main driving force for sintering. (Blackford, J. R. J. Phys. D: Appl. Phys. 40 (2007) R355–R385) In the case of frazil ice flocs in supercooled water, however, the frazil discs can simply freeze together due to the heat transfer from the boundaries of the frazil disks to the supercooled water. There is no need to look for a reduction in surface energy of the system to cause the disks to stick together. Also, it is well known that flocs form only in supercooled water. Ice crystals in slush, a mixture of ice and water all at the ice/water equilibrium temperature, do not stick together. Perhaps you are using the word "sintering" in a very general sense to describe solid particles sticking together without regard to the mechanism causing them to stick. That use is imprecise and confusing. The exact mechanism causing the frazil disks to fuse together should be described.

> We agree and replaced "sintering together" with "freezing together" everywhere.

Line 22. It is suggested that the authors consider providing more background on the process of floc formation. The frazil disks are transported by the flow. If the frazil disks are all moving at identical velocities, they cannot collide. Disk collisions require spatially varying disk velocities. Spatially varying disk velocities can result from spatially varying flow velocities and disk varying buoyant rise velocities. There are several mechanisms providing spatially varying flow including turbulent eddies of appropriate size and the influence of the stationary boundary at the channel bottom.

> We agree with this suggestion and revised the manuscript as follows (see L22-25):

> "As they are transported by the turbulent flow, they may collide with each other due to spatially varying particle velocities resulting from differential rising or due to spatially varying flow velocities created by turbulent eddies and boundary shear (Mercier, 1985). Colliding particles may freeze together forming clusters of particles known as frazil flocs in a process called flocculation (Clark and Doering, 2009)."

Line 25. It is suggested that the authors consider the vagueness of the term "grow." In the previous sentence you write: "Frazil flocs grow in size either by the thermal growth of the crystals and/or by further aggregation of individual frazil ice particles or flocs." Then you state "Once frazil flocs grow…" It seems to be that the word "grow" should be applied only to thermal growth of the crystals. Increase in size through aggregation is something different. Perhaps there can be two distinct types of growth, but you should make this clear.

> We agree and replaced "grow in size" with "increase in size" and revised the manuscript to only use "grow" when describing the thermal growth of crystals.

Line 41 (and other locations). It is suggested that the authors consider not using the terms "residual supercooling" and "principal supercooling" and replacing them with more accurate terms. According to the authors, frazil ice formation has two periods. The first is the "principal supercooling" period and the second, which follows the first, is the "residual supercooling" period. There is a long history of using the term "residual supercooling" going back to the very first experiments of Michel (Michel, Bernard. Properties and processes of river and lake ice. Université Laval, Laboratoire de mécanique des glaces, 1972.). However, the use of the term "residual" is very unsatisfactory. Residual describes what remains after most of something is gone. However, the supercooled temperature of the water is not a residual of the higher levels of supercooled water temperatures that were temporarily present during the earlier principal period of supercooling. The water temperature at all times represents a dynamic balance between the heat loss at the water surface and the latent heat released by the growing frazil ice is suspension and the anchor ice on the channel bed. The water temperature is more-or-less constant during the residual period because the heat loss at the water surface and the latent heat released by the

growing ice are equal. In summary, residual supercooling is not left over, it represents a dynamic heat balance exactly as in the principal period. The authors should consider replacing "principal supercooling" with "transient supercooling period" and "residual supercooling" with "steady-state supercooling period."

> This is an interesting suggestion supported by logical arguments. However, as noted the terms principal supercooling and residual supercooling have been in use for more than 50 years and have been used in numerous previous publications. Therefore, we decided to keep using the two conventional widely used terms because we think introducing new terminology will lead to confusion. We emphasized in the revised manuscript that residual supercooling occurs when a steady water temperature is reached, and that principal supercooling refers to the time period when the water temperature varies transiently, see L42-43.

Line 106. Table 1. It is suggested that the authors consider adding an additional term to their "Summary of the study reach characteristics" table. It is suggested that they add the term e, the turbulent energy dissipation rate per kilogram of fluid. This term strongly influences the heat transfer from suspended particles and the secondary nucleation rate. This can be estimated for both channel flow and laboratory tests. This parameter would allow the reader to compare field sites with previous laboratory tests. The units are generally in Wkg-1 with dimensions of m2s-3.

> We have estimated the turbulent dissipation rate from the bed slope and average width and depth listed in Table 1 and added this data to the table with descriptions of the data included in Sec. 2 (e.g., L86-88). The dissipation rate was 0.0058 and 0.0051 $m^2$ $s^{-3}$ in NSR and PR, respectively and was 0.2066 $m^2$ $s^{-3}$ in the small-steep mountain river KR. We also revised the discussion on floc sizes to include a comparison of the estimated dissipation rates and the dissipation rates measured in laboratory tank experiments that reported frazil floc measurements, see L482-483.

Line 117. Change "capture" to "image."

> Revised as recommended (L122).

Line 265. Change "4.2 Heat flux analysis" to "4.2 Heat flux analysis at the water surface"

> Revised as recommended (L283).

Line 265. Heat flux analysis. It is suggested that the authors verify the accuracy of their heat flux analysis at the water surface by modeling the water temperature decline early in the transient period prior to the formation of ice. This could be done for deployments NSR-L.1, NSR-L.3, and NSR-L.4. Two basic and reasonable assumptions would make the model simple and

straightforward: that there are no significant gradients of temperature in the longitudinal direction (parallel to the flow velocity) and that the water temperature was well mixed in the vertical direction.

Thank you for your suggestion. We did the suggested calculation for NSR-L3 and NSR-L4. NSR-L1 only captured the warming period of principal supercooling therefore cannot be used. Figure R1 shows an example of calculated and measured water temperature during NSR-L4. The calculated water temperature is consistently lower than the observed. The suggested method assumes no ice is forming and releasing latent heat, and that the water temperature is only affected by the air-water heat flux. However, as shown in Fig. R2 which is a game camera image captured during NSR-L4 near the deployment site, surface ice pans and border ice were observed while no suspended flocs were measured by FrazilCam. Pans, border ice and possibly skim ice may have been growing in the supercooled water and releasing latent heat into the water, thus by neglecting the growth of other ice in the river this method appears to overestimate the magnitude of the water temperature decline. In addition, as noted in the discussion section of the manuscript, the heat flux analysis did not account for the surface ice coverage, which may also contribute to the lower calculated water temperatures. We think this calculation will only provide realistic estimates at the very start of river freeze-up when there is no other significant ice formation occurring. Therefore, we concluded the suggested method could not be used to verify the heat flux analysis.

In a previous study the co-authors investigated various formulas used to calculate downwelling longwave radiation and the latent and sensible heat fluxes during freeze-up on the same reach of the North Saskatchewan River. Yang et al. (2023b) compared measured and *River1D* modeled water temperatures and determined which combination of formulas provided the most accurate results. This same combination of formulas was used in this study. Therefore, we are reasonably confident that the estimated heat fluxes are sufficiently accurate. We revised the heat flux analysis section to provide a clearer description and justification of the methods used for the heat flux analysis (L283-323).

[Figure]

Figure R1. Calculated and observed water temperature time series during NSR-L4.

[Figure]

Figure R2. A game camera image captured 1.5 km upstream of Laurier Park site at 3 pm during NSR-L4.

Line 394. Revise section starting with "Arakawa (1954) discovered …" and ending with "time to grow irregularly." (Line 398) It has long been realized that the stability of the edge of the ice crystals is controlled by the formation of temperature gradients in the water at the ice/water interface when the surrounding water is supercooled (Mullins, W. W. and R.F. Serkerka (1964) Stability of a planar interface during solidification of a dilute binary alloy. Journal of Applied Physics, 35, No. 7, 444-451). The perfect disk shape of frazil ice results from the anisotropic crystalline kinetics combined with the turbulent suppression of temperature gradients surrounding the crystals. Given the ability of turbulence to suppress gradients through mixing, unstable disk growth is typically a special case. Irregular particles generally indicate that the frazil ice particle has been in quiescent regions with exceptionally low turbulence levels. In these regions temperature gradients can form in the water surrounding the ice particle. Small perturbations of the ice crystal boundary encounter colder water because of the temperature gradients and grow more rapidly.

Thank you for providing this very helpful information. We revised this section as suggested, see L437-440; L445-454; and L476-478.

Line 385. 6. Discussion. It is suggested that the authors address these two related questions in this section. 1. How do you explain the near constant supercool water temperatures during the steady-state supercooling period based on your observations of suspended frazil disks and flocs? 2. What fraction of the total ice created in the water column is being sampled by the apparatus? The total ice created can be estimated based on the surface heat flux and the water temperature.

1. Constant water temperature during residual supercooling indicates that ice was still being formed and releasing latent heat that balances the heat loss from the surface. This could be due to various reasons. First, in our measurements of frazil flocs, fluctuations and trends in the floc number and volume concentration time series are observed although the mean floc size did not vary significantly during the residual supercooling period. This indicates that there may have been frazil ice particles still growing and forming flocs, releasing latent heat to help balance the surface heat loss. Secondly, as discussed below, suspended flocs comprised only a small fraction of the total ice in the river. During this period anchor ice, border ice, and surface ice pans were likely growing and releasing latent heat that balanced the surface heat loss. We added a paragraph in the discussion to address this topic, see L550-558.

2. We performed the suggested calculations for deployment NSR-L4. Time series of the observed floc volume concentration and calculated ice concentration are compared in Figure R3a. This data shows that the FrazilCam was only sampling up to 2% of the total ice that was forming in the water. It should be noted that this calculation involves significant approximations and assumptions. For example, the net heat flux used in the calculation does not account for the effect of surface ice due to a lack of accurate surface ice data. In addition, mean water depth was used while in reality water depth varied spatially and temporally. This introduce errors in the calculation of the total heat loss from the water surface, and the calculation of the volume of the water being cooled. Therefore, the accuracy of the calculated concentration is difficult to estimate precisely, but the uncertainty is likely not greater than a factor of two of three. Therefore, despite this potential large uncertainty, the calculations suggest that the FrazilCam was only sampling less than ~5% of the total ice being formed in the river. We added this analysis to the discussion section, see L560-575.

We also performed the suggested calculations using data from a recent laboratory frazil ice tank experiment measured by a lab version of the FrazilCam, and the results are shown in Fig. R3b. In the laboratory environment, the water depth is a constant and the surface heat loss can be quantified from the water cooling rate with reasonable accuracy. The results show that the calculated concentration started rising earlier than the measured

suspended floc concentration, which is possibly because the measured data did not include the frazil particles and surface skim ice. The trend in the observed and measured concentrations aligned quite well between 2050 and 2170 s where the calculated and measured time series are overlapping. After that the measured data decreased due to flocs rising to the surface while the calculated time series was still increasing since the calculation does not account for the rising of flocs. Overall, the alignment between the calculated and observed time series prior to the rising of flocs demonstrates that the FrazilCam does provide accurate measurements of the suspended ice concentration. This also suggests that the only time the FrazilCam would be sampling a significant fraction of the total ice being formed in the river would be when suspended frazil is the only ice that is actively growing. We added a brief description of this analysis on L575-582.

[Figure]

Figure R3. Calculated total ice concentration $C_i$ and observed suspended frazil floc concentration $C_{fv}$ during (a) NSR-L4 and (b) a lab frazil tank experiment using a similar apparatus.

**Response to Referee #2**

General comments and suggestions:

The authors present a well written study which investigates the properties and concentrations of frazil flocs in reaches of the North Saskatchewan River (NSR), Peace River (PR), and Kananaskis Rivers (KR) using a submersible camera deployment called FrazilCam. The manuscript presents in-situ observations of floc shape, size, and concentration which is of use to the frazil and river ice communities for improved modelling efforts. The following general suggestions can be considered by the authors for potentially improving the quality of the manuscript. Specific comments follow these general suggestions.

1. The methodology used for classification of flocs was challenging to follow at times. Perhaps adding in a flowchart or process-flow diagram would aid in reader comprehension to better follow all the processing steps.

We agree and added a flowchart (Fig. 6, L220-221) to the revised manuscript.

2. The heat flux analysis requires additional clarification and minor reanalysis. Namely, description of equations used and some minor addendums to methodology. Please see specific comments.

We revised Sec. 4.2 Heat flux analysis at the water surface to include a description of equations used and justifications for why we chose these equations, see L283-323.

Detailed comments and suggestions:

Study Reaches:

L108-111: Is the data obtained from Kellerhals et al. (1972) the most up to date site data available?

Yes. To our knowledge it is up to date and is representative of the current characteristics of the study reaches.

L114: The site map is well made and clear.

Thanks!

Instrumentation, Methodology, and Deployments:

L115 and L126: The study modifies the FrazilCam system developed by MacFarlane et al. (2017). Can the specific modifications from the original system be made more clear in this section? Was the FOV the only aspect modified (L126-127).

The system was modified to enlarge the FOV and increase the gap between the polarizers. We revised L125-137 to make this clearer.

L147: Which frequency Aquadropp ADCP was used? My concern is for blanking distances, as the river sites have quite shallow depths (Table 3).

We used a 2 MHz AquaDopp High Resolution Profiler with a blanking distance of 0.1 m. As mentioned in L152, we only used the velocities measured at 60% of the water depth to get estimates of depth-averaged water velocity so the blanking distance should not be an issue. We revised L151-155 to make these details clearer.

Image Processing:

L221-223: Were the preliminary experiments conducted by yourself? If not, please provide some more context.

*Yes, the preliminary experiments were conducted by myself.*

L234: 'S' is not defined prior to its use here.

*S was previously defined at L224.*

L237: Perhaps it should be made clearer earlier in the study that only 1 deployment coincided with an entire 'principal' supercooling event.

*We agree and added a sentence in Sec. 3 right before Table 3 to make this clear earlier, see L188-190.*

L231-264: The addition of a process-flow diagram or flowchart would largely help the reader understand the methodology used. Additionally, it would be useful to consider provided quantitative measurements of how many images were taken (in total), followed by how many were removed at each processing step.

*The requested diagram was added as Fig. 6 and Table 3 was revised to add number of images processed for each deployment as suggested. We also added that images removed during the rinsing period constituted 2~14% of the total number of images, see L212-213.*

L255: One key missing piece of information was the specific sampling time used for each deployment. On L131, it is noted that 5 images at 1 Hz every 9,15, or 19s were acquired depending on field conditions. It would be more transparent to describe under what case/deployment each sampling time was used. If sampling times were mixed for a given deployment, this should be also stated and justified.

*We agree and revised Section 3 right before Table 3 (L179-181) to provide the following information: NSR-L1 to L6 and KR-E1 used 5 images every 9s; PR-F1 and F2 used 5 images every 18s; and KR-F1 and F2 used 5 images every 15s, sampling times were not mixed for any given single deployment. Note that, we noticed a typo in the manuscript - the third time interval is 5 images at 1 Hz every 18s, not 19s and this error was corrected in the revised manuscript (L139-140).*

Heat Flux Analysis:

L273: It is uncommon in heat flux analysis to explicitly consider an albedo in the longwave spectrum as nearly all radiation in this spectrum is thought to be absorbed at/near the surface (shown by your use of a very small albedo of 0.03). It is recommended to remove this.

The albedo of 0.03 in the longwave radiation calculation was calculated for water temperatures of normal ranges (Raphael, 1962). It has been used in multiple recent river energy budget studies in the river ice fields (e.g., Richard et al., 2015; McFarlane and Clark, 2021; Yang et al. 2023a). Therefore, we decided to keep the albedo of 0.03 to be consistent with these studies.

L277: Mean water temperatures are used here for conducting a surface energy balance. While vertical turbulence may be well developed and river depths relatively shallow, some degree of caution should be presented on this matter. Surface temperatures may deviate significantly depending on flow and meteorological conditions. The assumption that the river reach is fully vertically mixed should be stated explicitly in this case.

We agree and stated this assumption explicitly at L295-297 as recommended.

L278-280: Satterlund (1979)'s parameterization relies on data from Aise and Idso (1978) from continental Montanna and is extended with Stoll and Hardy (1955) for measurements in Alaska. Perhaps it may be more prudent to select a more well-used scheme for clear sky conditions shown effective in higher latitude regions of North America (e.g. Efimova, 1961). Key et al. 1996 provide a review on the matter using data from Alaska and the Northwest Territories on their review of parameterization schemes. It is left to the authors' discretions to keep the current scheme or adopt one of the above-mentioned after reviewing the noted references.

Efimova, N. A. (1961). On methods of calculating monthly values of net longwave radiation. Meteorol. Gidrol., 10, 28-33

Key, J.R., Silcox, R.A., Stone, R.S., 1996. Evaluation of surface radiative flux parameterizations for use in sea-ice model. J.Geophys. Res. 101 C2, 3839–3849

Thank you for this suggestion. In a previous study (Yang et al., 2023b), the co-authors investigated various formulas used to calculate downwelling longwave radiation and the latent and sensible heat fluxes during freeze-up on the North Saskatchewan River. Formulas for the downwelling longwave radiation for clear-sky atmospheric emissivity and cloud effect were assessed using direct measurements. The combination of formulas used in this study were the ones that provided the most accurate results in Yang et al (2023b). Therefore, we decided to keep the current scheme. We have revised the manuscript to justify the current scheme in L317-323.

L281: Where was Bowen's ratio obtained from? In addition to the above-mentioned, please describe equations used for the flux analysis within this section. It provides the reader with the information readily, rather than having to access several other sources to understand the approach taken.

The Bowen's ratio is given by:

$$B = \frac{C_a P}{0.622 l_v} \times \frac{T_s - T_a}{e_s - e_a},$$

where $C_a$ is the specific heat of air; $l_v$ is the latent heat of vaporization; $P$ is the atmosphere pressure; $T_s$ is the surface water temperature taken the same as the measured water temperature. $T_a$ is the air temperature; $e_s$ and $e_a$ are the saturated and actual vapour pressure of water, respectively.

We revised this section as suggested to include all the equations used for the heat flux analysis as well as justifications for why we chose these equations, see L283-323.

Results:

Floc Shape, Size and Concentration:

L303: It was quite interesting to record such a large floc size (99.69mm). As I understood, the FrazilCam in this study had an increased FOV relative to its predecessor. Would recording a max floc size such as this in KR-E1 suggest perhaps the FOV may need to be further increased? Perhaps there may be potential for biasing floc sizes too low, as larger flocs that are unfavourably oriented interact with the polarizers and break.

Yes, this does suggest that further increases in both polarizer size and the gap between the polarizers might be needed to allow even larger flocs to be imaged. We added a short discussion to the revised manuscript regarding the possibility that the current system may be under-sampling the large flocs, see L489-493.

Floc Size Distribution:

L319: You note that lognormal distributions are reasonable fits for the distributions. Would you be able to provide a quantitative measure of the fit for each of the histograms?

This statement was based only on visual examination that a theoretical lognormal distribution was a reasonable fit to the observed distributions. As suggested we attempted to provide quantitative evidence for this statement. We applied the Chi-Squared test and none of the distributions passed the test with a 5% significance level. This could be due to various reasons. First, as discussed in Sec. 5.2, the use of cut-off size to eliminate sediment particles produces a sharp cut-off in the distributions, the effect is more pronounced in deployments that recorded smaller floc sizes. Second, the small number of samples in some deployments resulted in noisy size distributions making it unlikely that they would be a good quantitative fit for a smooth lognormal distribution. We revised the discussion section to make clear that our conclusion that the measured distributions are approximately lognormal is based on visual comparisons and that the Chi-Squared test did not statistically confirm the lognormal distribution, see L495-503.

Time Series:

L332: It is understandable that not all sites are presented within the contents of the manuscript. I do however believe that the reader is left curious as what the other sites might have looked like. Perhaps it can be considered to add in the other deployments data (similar or simpler versions to Figure 8,9,10) as supplementary data for further transparency.

> This is a good suggestion and we provided the time series from other deployments as supplementary data as recommended.

L357-358: The usage of hourly component flux data for correlating floc properties and concentrations is a bit questionable given the timescales of the deployments (~1-3hr). It is described between L155-167 where this data is obtained from. If these sites (minus the NSR reach) have sub-hourly data that can be used for the heat flux analysis, please consider updating.

> Unfortunately, we do not have sub-hourly data for the KR and PR sites to update our results. Please note that the hourly heat fluxes were plotted in the time series only to illustrate the trends of the heat flux variations and to identify the dominant component during the deployments. We think this is useful because it facilitates better understanding of supercooling which is directly related to the surface heat flux. We did not try to correlate the hourly heat flux time series data with the floc properties or concentration time series data.

L375, 380, 383: Figures 8,9, and 10 would benefit from the addition of air temperature data.

> We agree and added the air temperature data as recommended.

Discussion:

L483-484: A larger limitation would be the use of hourly data rather than neglecting surface conditions and sediment heat fluxes. This is likely too coarse for the intended goal with correlating net surface heat flux with floc properties. This can be considered going forward for future studies on the matter.

> We agree that this is a limitation of the study and we hope in the future to deploy our own weather stations to obtain meteorological data at faster sampling rates at these more remote sites. We did conduct a correlation analysis using only the heat flux data with 10-min resolution from the six NSR deployments and still did not find any significant correlations. Therefore, including the deployments with the 1-hour heat flux data did not change the current result. We revised L589-593 to describe this analysis using only the 10-min data and explain that there was still no correlations found in the revised manuscript.

**Response to the Editor Comments**

1) Please update the list of references to fit TC requirements.

Updated as suggested.

2) The heat flux analyses part is weak. Please consider using 1minute data for the analyses.

Unfortunately, we do not have 1-min data to update our results. Among the eleven deployments, the heat fluxes in all six NSR deployments were calculated on a 10-min time interval because we had a weather station and net radiometer deployed close to the river bank. For the remaining five deployments in the remote KR and PR regions, the heat fluxes were calculated on a 1-hour time interval due to lack of available data. We hope in the future to deploy our own weather stations to obtain meteorological data at faster sampling rates at these more remote sites. We revised the manuscript (L283-323) to make the description, limitations and justifications of the heat flux analysis clearer.

3) I suggest authors describe the possible application of FrazilCam system for lake ice and sea ice in situ observations.

We added a paragraph at the end of the manuscript to discuss the possible application of FrazilCam for lake and ocean deployments, please see L661-666.

4) I recommend having to deploy an eddy flux sonic anemometer to measure turbulent fluxes in future observations.

Thanks for the recommendation. We will explore the feasibility of incorporating an eddy flux sonic anemometer in our future deployments.

5) As an option, I think this study would become much stronger if authors could give an estimation on how frazil floc distributions can be linked with river ice freezing up, e.g. initial ice freezing-up date.

As discussed in the introduction section, the formation and evolution of frazil flocs in rivers starts as soon as frazil ice particles are formed in the supercooled water at the beginning of the river freeze-up. Flocs will rise to the surface once they gain enough mass and form frazil pans at the surface. This process largely determines the development of a solid ice cover until the entire river section is covered with ice and supercooling is stopped. The floc distributions alone cannot be linked with the final ice cover formation directly since flocs are being actively produced as long as the water is supercooled. However, the river freeze-up process can be modelled to predict the ice cover development. Current comprehensive river ice models like *River1D* did not explicitly model the frazil flocculation and floc evolution due to a lack of data and understanding of the flocculation and floc properties. Our measurements could facilitate the improvements of the river ice models to better predict the freeze-up processes that lead to the ice cover formation. We revised the conclusion section at L652 and L656-659 to make this clearer.

References

Clark, S. P. and Doering, J. C.: Frazil flocculation and secondary nucleation in a counterrotating flume, Cold Reg. Sci. Technol., 55(2), 221-229, https://doi.org/10.1016/j.coldregions.2008.04.002, 2009.

McFarlane, V. and Clark, S.P.: A detailed energy budget analysis of river supercooling and the importance of accurately quantifying net radiation to predict ice formation, Hydrol. Processes, 35(3), e14056, https://doi.org/10.1002/hyp.14056, 2021.

McFarlane, V., Loewen, M. and Hicks, F.: Measurements of the evolution of frazil ice particle size distributions, Cold Reg. Sci. Technol., 120, 45-55, https://doi.org/10.1016/j.coldregions.2015.09.001, 2015.

McFarlane, V., Loewen, M. and Hicks, F.: Measurements of the size distribution of frazil ice particles in three Alberta rivers, Cold Reg. Sci. Technol., 142, 100–117, https://doi.org/10.1016/j.coldregions.2017.08.001, 2017.

Mercier, R. S.: The reactive transport of suspended particles: mechanisms and modeling, Ph.D. thesis, Massachusetts Institute of Technology, United States, 1985.

Raphael, J.M.: Prediction of temperature in rivers and reservoirs, J. Power Div. 88, 157–181, https://doi.org/10.1061/JPWEAM.0000338, 1962.

Richard, M., Morse, B. and Daly, SF.: Modeling frazil ice growth in the St. Lawrence River, Can. J. Civ. Eng., 42 (9), 592–608, https://doi.org/10.1139/cjce-2014-0082, 2015.

Yang, J., She, Y. and Loewen, M.: Assessing heat flux formulas used in the full energy budget model for rivers during freeze-up, in: CGU-HS Committee on River Ice Processes and the Environment (CRIPE) Proceedings of the 22nd Workshop on the Hydraulics of Ice Covered Rivers. Canmore, Canada. 9-12 July 2023, 2023b.

Yang, J., She, Y. and Loewen, M.: Assessing the uncertainties in modeling water temperatures during river cooling and freeze-up periods, , Cold Reg. Sci. Technol., 210, 103840, https://doi.org/10.1016/j.coldregions.2023.103840, 2023a.